# On the Throughput of the Common Target Area for Robotic Swarm Strategies

**Yuri Tavares dos Passos** [1,2,*] **, Xavier Duquesne** [2] **and Leandro Soriano Marcolino** [2]

1 Centro de Ciências Exatas e Tecnológicas, Universidade Federal do Reconcâvo da Bahia, Rua Rui Barbosa, 710. Centro., Cruz das Almas 44380-000, Brazil

2 School of Computing and Communications, Lancaster University, Bailrigg, Lancaster LA1 4WA, UK; duquesne.xavier.13@gmail.com (X.D.); l.marcolino@lancaster.ac.uk (L.S.M.)

* Correspondence: yuri.passos@ufrb.edu.br

**Abstract:** A robotic swarm may encounter traffic congestion when many robots simultaneously attempt to reach the same area. This work proposes two measures for evaluating the access efficiency of a common target area as the number of robots in the swarm rises: the maximum target area throughput and its maximum asymptotic throughput. Both are always finite as the number of robots grows, in contrast to the arrival time at the target per number of robots that tends to infinity. Using them, one can analytically compare the effectiveness of different algorithms. In particular, three different theoretical strategies proposed and formally evaluated for reaching a circular target area: (i) forming parallel queues towards the target area, (ii) forming a hexagonal packing through a corridor going to the target, and (iii) making multiple curved trajectories towards the boundary of the target area. The maximum throughput and the maximum asymptotic throughput (or bounds for it) for these strategies are calculated, and these results are corroborated by simulations. The key contribution is not the proposal of new algorithms to alleviate congestion but a fundamental theoretical study of the congestion problem in swarm robotics when the target area is shared.

**Keywords:** robotic swarm; common target; throughput; congestion; traffic control

**MSC:** 68T40; 70-10

## 1. Introduction

Swarms of robots are systems composed of a large number of robots that can only interact with direct neighbours and follow simple algorithms. Interestingly, complex behaviours may emerge from such straightforward rules [1,2]. An advantage of such systems is the usage of low-priced robots instead of a few expensive ones to solve problems. Robotic swarms accurately projected for simple robots may solve complex tasks with greater efficiency and fault-tolerance, while being cheaper than a small group of complex robots oriented for a specific problem domain. They can also be seen as a multi-agent system with spatial computers, which is a group of devices displaced in the space such that its objective is defined in terms of spatial structure and its interaction depends on the distance between them [3]. Swarms have recently been receiving attention in the multi-agents systems literature in problems such as logistics [4], flocking formation [5], pattern formation [6] and the coordination of unmanned aerial vehicle swarms [7]. In such problems relating to spatial distribution, conflicts may be created by the trajectories of the robots, which may slow down the system, especially when a group is intended to go to a common region of the space. Some examples where this happens are waypoint navigation [8] and foraging [9].

The topic of robotic traffic control has been studied for a long time [10–12], but with the premise that autonomous cars navigate on delimited lanes and that coordination is needed only at junctions. Even recent related works on multi-agent systems [13–15] also

deal with this problem in a similar way. In [16,17], they also deal with multi-agents and pathfinding, but not in a situation where the target of every agent is the same area. Furthermore, distributed solutions are considered here where agents only have local information, while [16,17] propose centralised solutions. Xia et al. [18] investigate the topology of the neighbourhood relations between multiple unmanned surface vehicles in a swarm. They deal with maintaining formation in swarms, but they have to keep virtual leaders, and their goal is not to minimise congestion.

Moreover, there has not been much research on the problem of reducing congestion when a swarm of robots is aimed at the same target. Surveys about robotic swarms [19–24] do not provide information regarding these situations. Even a recent survey on collision avoidance [25] does not address this issue, though it provides insights into multi-vehicle navigation. Congestion in robotic swarms is mostly managed by collision avoidance in a decentralised fashion, allowing for improved algorithm scalability.

However, solely avoiding collisions does not necessarily lead to a good performance in problems with a common target. For example, Marcolino et al. [26] showed that the ORCA algorithm [27] reaches an equilibrium where robots could not arrive at the target despite avoiding collisions. That paper also presented three algorithms using artificial potential fields for the common target congestion problem, but no formal analysis of the cluttered environment was conducted. Hence, congestion is still not well understood, and more theoretical work is needed to measure the optimality of the algorithms. A better understanding of this topic should lead to a variety of new algorithms adapted to specific environments. Thus, this paper aims to introduce the first theoretical study on this problem, which should lead to future enhancements in handling congestion in robotic swarms.

Therefore, this work fits in the literature on mathematical models of swarm robotics, such as the works by Lima and Oliveira [28], which models a cellular automata ant memory to control a robot swarm for foraging tasks; Varghese and McKee [29], for pattern transformation modelling; Li and Chen [30], for box-pushing; Taylor-King et al. [31], which studies the effect of turning delays on the behaviour of groups of robots; Galstyan et al. [32], for microscopic robots that reside in a fluid and can detect chemicals; Khaluf and Dorigo [33], which models swarm performance measures using the integral of linear birth–death processes; and Mannone et al. [34], which uses category theory and quantum computing to model the development of robotic swarm systems. However, as mentioned, these theories do not yet allow one to better understand swarm congestion.

Furthermore, any elaborated analysis on that subject must investigate the effect of the increase in the number of individuals on the swarm congestion, as it is desirable for the system to perform well as it grows in size. If one has a finite measure that abstracts the optimality of any algorithm as the number of robots goes to infinity, this can be used as a metric to compare different approaches to the same problem. Thus, this work presents as a metric the common target area throughput. That is, a measure of the rate of arrival in this area is proposed as the time tends to infinity as an alternative approach to analyse the congestion in swarms with a common target area. In network and parallel computing studies [35,36], asymptotic throughput is used to measure the throughput when the message size is assumed to have infinite length. The same idea is used here, but instead of message size, it is applied with infinite time, as if the algorithms run forever. As it will be presented in the next section, this implies dealing with an infinite number of robots. Thus, time is being used here instead of message size or bytes, as in computer network studies.

Therefore, the contributions in this paper are the following.

1. A method for evaluating algorithms for the common target problem in a robotic swarm by using the throughput in theoretical or experimental scenarios is proposed.
2. An extensive theoretical study of the common target problem is presented, allowing one to better understand how to measure the access to a common target using a metric not yet used in other works on the same problem.
3. Assuming a circular target area and that the robots are constantly moving at the maximum linear speed and have a fixed minimum distance from each other, theo-

retical strategies for entering the area are developed, and their maximum theoretical throughput for a fixed time and their maximum asymptotic throughput when time goes to infinity are calculated (or bounds for it). Additionally, the correctness of these calculations is verified by simulations.

The presented theoretical strategies are based on forming a corridor towards the target area or making multiple curved trajectories towards the boundary of the target area. For the corridor strategy, the throughput when the robots are moving towards the target in square and hexagonal packing formations is also discussed. The theoretical strategies are evaluated by realistic Stage [37] simulations with holonomic and non-holonomic robots. These experiments corroborate that whenever an algorithm makes a swarm take less time to reach the target region than another algorithm, the throughput of the former is higher than the latter.

Note that the key contribution of this work is not the proposal of new algorithms to alleviate congestion but a fundamental theoretical study of the congestion problem in swarms having the same target. The presented strategies are the theoretical grounding for new distributed algorithms for robotic swarms in our concurrent work [38]. When we assume that the robots are constantly moving at maximum linear speed and maintaining a fixed minimum distance, we can provide analytical calculations of the maximum possible throughput for a given time and bounds or exact value of the maximum asymptotic throughput for the different theoretical strategies. Based solely on these calculations, we can compare which strategy is better. However, for robots using artificial potential fields, it is not straightforward to obtain explicit throughput equations due to the changeability of those quantities previously assumed constant. Then, in the lack of closed asymptotic equations, simulations were performed in [38] for the algorithms inspired by our strategies in order to obtain experimental throughput and compare algorithms for varying linear speeds and inter-robot distances. As shown by these experimental data, their variation and the effect of the other robots in the trajectory does affect the throughput. However, the analytically calculated maximum throughput in this work serves as an upper bound to the ones obtained from the simulations in more realistic conditions when considering the mean speed and mean distance between the robots in place of the constant values on the obtained equations.

This paper is organised as follows. The next section briefly explains the mathematical notation being used. Section 3 formally defines the common target area throughput and proves statements about this measure for theoretical strategies that allow robots to enter the common target area. Section 4 describes the experiments and presents their results to verify the correctness of the theoretical strategies results. Finally, Section 5 summarises the results and gives final remarks.

## 2. Notation

Geometric notation is used as follows. $\overleftrightarrow{AB}$, $\overrightarrow{AB}$ and $\overline{AB}$ represent a line passing through points A and B, a ray starting at A and passing through B and a segment from A to B, respectively. $|\overline{AB}|$ is the size of $\overline{AB}$. $\overleftrightarrow{AB} \parallel \overleftrightarrow{CD}$ means $\overleftrightarrow{AB}$ is parallel to $\overleftrightarrow{CD}$. If a two-dimensional point is represented by a vector $P_1$, its x- and y-coordinates are denoted by $P_{1,x}$ and $P_{1,y}$, respectively.

$\triangle ABC$ expresses the triangle formed by the points A, B and C. $\triangle ABC \cong \triangle DEF$ and $\triangle ABC \sim \triangle DEF$ mean the triangles ABC and DEF are congruent (same angles and same size) and similar (same angles), respectively. Depending on the context, the notation is omitted for brevity.

$\widehat{AOB}$ means an angle with vertex O, one ray passing through point A and another through B. Depending on the context, if only one $\triangle EFG$ is being dealt with, its angles will be named only by $\widehat{E}$, $\widehat{F}$ and $\widehat{G}$. All angles are measured in radians in this paper.

## 3. Theoretical Analysis

This paper considers the scenario where a large number of robots must reach a common target. After reaching the target, each robot moves towards another destination which may or may not be common among the robots. It is assumed that the target is defined by a circular area of radius $s$. A robot reaches the target if its centre of mass is at a distance below or equal to the radius $s$ from the centre of the target. In addition, it is supposed that there is no minimum amount of time to stay at the target. Additionally, the angle and the speed of arrival have no impact on whether the robot reached the target or not. In this section, theoretical strategies are constructed to solve that task and show limits for the efficiency of real-life implementations, which we developed in a concurrent work [38]. To measure performance, the following definition is presented.

**Definiton 1.** *The* throughput *is the inverse of the average time between arrivals at the target.*

Informally speaking, the throughput is measured by someone located on the common target (i.e., on its perspective). It is considered that an optimal algorithm minimises the average time between two arrivals or, equivalently, maximises throughput. The unit for throughput can be in $s^{-1}$. It will be noted $f$ (as in frequency). The rest of the paper focuses on maximising throughput.

Assume an experiment was run with $N \geq 2$ robots for $T$ units of time, such that the time between the arrival of the $i$-th robot and the $i+1$-th robot is $t_i$, for $i$ from 1 to $N-1$. Then, by Definition 1, $f = \frac{1}{\frac{1}{N-1}\sum_{i=1}^{N-1} t_i} = \frac{N-1}{\sum_{i=1}^{N-1} t_i} = \frac{N-1}{T}$, because $\sum_{i=1}^{N-1} t_i = T$. Thus, an equivalent definition of throughput is given:

**Definiton 2.** *The* throughput *is the ratio of the number of robots that arrive at a target region, not counting the first robot to reach it, to the arrival time of the last robot.*

The target area is a limited resource that must be shared between the robots. Since the linear speeds of the robots have an upper bound, a robot needs a minimum amount of time to reach and leave the target before letting another robot in. Let the *asymptotic throughput* of the target area be its throughput as the time tends to infinity. Because any physical phenomenon is limited by the speed of light, this measure is bounded. Then, the asymptotic throughput is well suited to measure the access of a common target area as the number of robots grows.

One should expect that the asymptotic throughput depends mainly on the target size and shape, the speed of the robots, and the distance between robots. As any bounded target region can be included in a circle of radius $s$, only circular target regions will be dealt with hereafter. If the robots are moving at maximum speed and keeping the distance between each at a minimum value all the time, then it is also expected that the throughput and asymptotic throughput reach their maximum value. Thus, it is assumed hereafter that the robots move at a constant maximum linear speed, $v$, and the distance between each other is either constant when possible or no lower than a fixed value, $d$.

To efficiently access the target area, two main cases are identified: $s \geq d/2$ and $s < d/2$. There are targets that several robots can simultaneously reach without collisions. That is the case if the radius $s \geq d/2$. Thus, one approach is making lanes arrive in the target region so that as many robots as possible can simultaneously arrive. After the robots arrive at the target, they must leave the target region by making curves. However, we discovered [38] that this approach does not obtain good results in realistic simulations due to the influence of other robots, although it is theoretically the best approach if the robots could run at a constant speed and maintain a fixed minimum distance between each other.

The case where $s < d/2$, when only one robot can occupy the target area simultaneously, is of interest. Making two queues and avoiding the inter-robot distance being less than $d$ is good guidance to work efficiently. Particularly, the case $s = 0$ offers interesting insights, so this is discussed next.

Some lemmas and propositions need a long technical treatment to be proven. In order to avoid the reader missing the main idea of this paper, only their statements are provided. All proofs are available in the Supplementary Materials.

*3.1. Common Target Point: $s = 0$*

Consider the case where robots are moving in straight lines at constant linear speed $v$, maintaining a distance of at least $d$ between each other. A robot has reached the target when its centre of mass is over the target. When $s = 0$, the target is a point. The first result is the optimal throughput when robots are moving in a straight line to a target point. It is illustrated in Figure 1. This section constructs a solution to attain the optimal throughput.

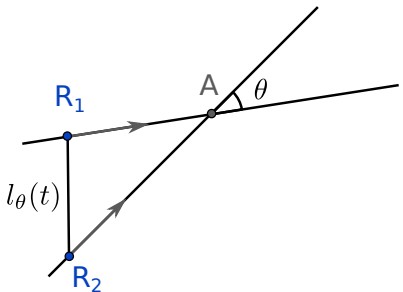

**Figure 1.** Two robots, $R_1$ and $R_2$, are moving in straight lines toward a target at A. The angle between their trajectory is $\theta$. The distance between the two robots over time is denoted by $l_\theta(t)$.

First, consider two robots, Robot 1 and Robot 2. Their trajectories are straight lines towards the target. Assume the straight-line trajectory of Robot 1 has an angle $\theta_1$ with the $x$-axis and the one of Robot 2 has $\theta_2$. Define $\theta_2 - \theta_1 = \theta$ as the angle between the two lines. The positions of the robots are described by the kinematic Equation (1) below, where $(x_1(t), y_1(t))$ and $(x_2(t), y_2(t))$ are the positions of Robot 1 and Robot 2, respectively, and $t \in \mathbb{R}$ is an instant of time. Without loss of generality, the origin of time is set when Robot 1 reaches the target, and the target is located at $(0,0)$. Thus, $(x_1(0), y_1(0)) = (0,0)$. $\tau$ is the delay between the two arrivals at the target. Then, $(x_2(\tau), y_2(\tau)) = (0,0)$, and

$$\begin{bmatrix} x_1(t) \\ y_1(t) \end{bmatrix} = \begin{bmatrix} vt\cos(\theta_1) \\ vt\sin(\theta_1) \end{bmatrix} \text{ and } \begin{bmatrix} x_2(t) \\ y_2(t) \end{bmatrix} = \begin{bmatrix} v(t-\tau)\cos(\theta_2) \\ v(t-\tau)\sin(\theta_2) \end{bmatrix} \tag{1}$$

In order to find the optimal throughput, this paper starts with its first lemma:

**Lemma 1.** *To respect a distance of at least d between the two robots, the minimum delay between their arrival is $\frac{d}{v}\sqrt{\frac{2}{1+\cos(\theta)}}$.*

This result leads to Proposition 1.

**Proposition 1.** *The optimal throughput $f$ for a point-like target ($s = 0$) is $f = \frac{v}{d}$. It is achieved when robots form a single line, i.e., the angle between the trajectories of the robots must be 0.*

The insight derived from Proposition 1 implies that one should increase the speed of the robots or decrease the minimum distance between them to increase the throughput. It is also noted that the optimal trajectory for all the robots is to form a queue behind the target and Robot 1. As a result, the optimal path is to create one lane to reach the target. When the angle $\theta$ between the path of a robot and the next one is increased, a delay from the optimal throughput is introduced. For instance, Figure 2 shows the normalised delay for different angles $\theta$ (normalised by dividing $\tau$ by $\tau_{min} = d/v$) between two robots, according to Lemma 1. This figure shows that for an angle of $\pi/3$, the minimum delay is 15% higher than for an angle of 0, and the minimum delay is 41% higher for an angle of $\pi/2$.

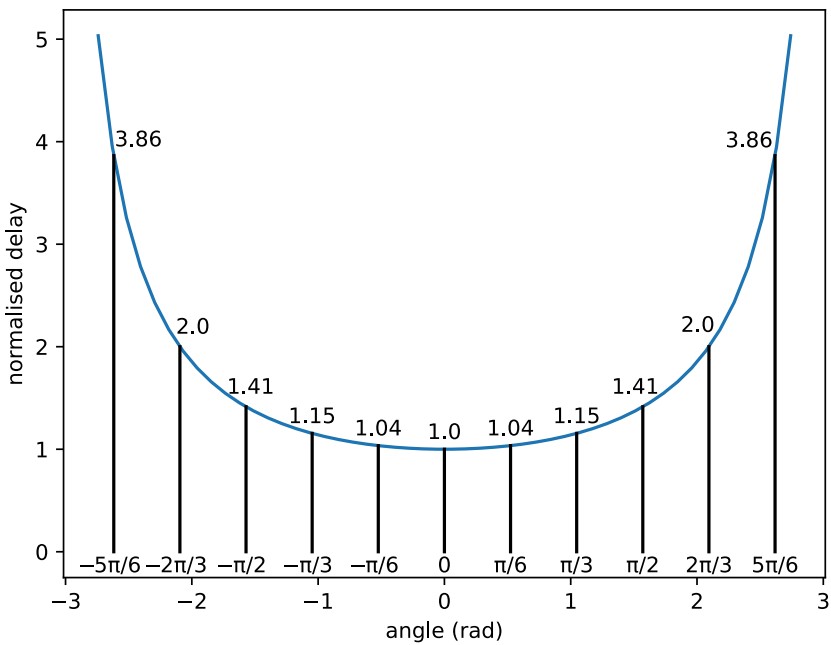

**Figure 2.** Normalised delay versus the angle between the trajectories of the robots.

### 3.2. Small Target Area: $0 < s < d/2$

This section supposes a small target area where $0 < s < d/2$; hence, two lanes with a distance $d$ cannot fit towards the target yet. The next results are based on a strategy using two *parallel lanes* as close as possible to guarantee the minimum distance $d$ between robots. Figure 3 describes these two parallel lanes. This strategy is called *compact lanes* hereafter. Proposition 2 considers a target area with radius $0 < s \leq \frac{\sqrt{3}}{4}d$, and Proposition 3 assumes $\frac{\sqrt{3}}{4}d < s < \frac{d}{2}$.

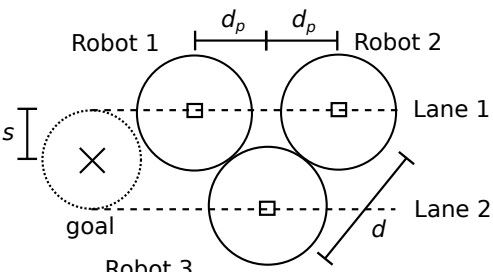

**Figure 3.** Two parallel robot lanes for a small target, illustrating the compact lanes strategy.

**Proposition 2.** *Assume two parallel lanes with robots at constant speed $v$ and maintaining a constant distance $d$ between them. The throughput of a common target area with radius $0 < s \leq \frac{\sqrt{3}}{4}d$ at a given time $T$ after the first robot has reached the target area is*

$$f(T) = \frac{1}{T}\left(\left\lfloor \frac{vT}{2\sqrt{d^2 - (2s)^2}} \right\rfloor + \left\lfloor \frac{vT}{2\sqrt{d^2 - (2s)^2}} + \frac{1}{2} \right\rfloor\right) \tag{2}$$

*and is limited by*

$$f = \lim_{T \to \infty} f(T) = \frac{v}{d\sqrt{1 - (\frac{2s}{d})^2}}. \tag{3}$$

**Proposition 3.** *Assume two parallel lanes with robots at constant speed $v$ and maintaining a constant distance $d$ between them. The throughput of a common target area with radius $\frac{\sqrt{3}}{4}d < s < \frac{d}{2}$ at a given time $T$ after the first robot has reached the target area is*

$$f(T) = \frac{1}{T}\left( \left\lfloor \frac{vT}{d} \right\rfloor + \left\lfloor \frac{vT}{d} + \frac{1}{2} \right\rfloor \right) \tag{4}$$

*and is limited by*

$$f = \lim_{T \to \infty} f(T) = \frac{2v}{d}. \tag{5}$$

Observe that if $T = k\frac{d}{v}$ for any $0 < k \in \mathbb{Z}$ is used in (4), the compact lanes strategy can achieve the throughput of two parallel lanes of robots going in the direction of the target region when $T = k\frac{d}{v}$ for any $k \in \mathbb{Z}$ or when $T \to \infty$, even though two robots cannot reach the target region at the same time.

*3.3. Large Target Area: $s \geq d/2$*

This section focuses on situations where more than two robots can simultaneously touch the target. Three feasible strategies are presented.

The simplest strategy is to consider several parallel lanes being at a distance $d$ from each other. However, it is possible to obtain higher throughput. In particular, two other strategies are identified: (a) using parallel straight line lanes that may be distanced lower than $d$ and (b) robots moving towards the target following curved trajectories. Strategy (a) uses more than two compact lanes, extending the strategy presented in the previous section. By doing this, the robots fit in a hexagonal packing arrangement moving toward the target region. Strategy (b) uses a touch and run approach. In it, robots do not cross the target area, they only reach it and return in the opposite direction using curved trajectories which respect the minimum distance $d$.

The next section starts with the parallel lanes strategy, which has the lowest asymptotic throughput over the strategies presented in this section, for comparison with the other strategies. In particular, it will be used later as a justification for the lowest number of lanes used in the strategy (b) in (14) in Proposition 7. Following their description and properties, a discussion comparing them is provided.

3.3.1. Parallel Lanes

It is considered here that the robots are moving inside lanes. The lanes are straight lines, and the linear speed $v$ of the robots is constant. The lanes are separated by a distance $d$, and each robot maintains a distance $d$ from each other. Figure 4 illustrates an example of this strategy. The first lane, Lane 1, is at the top. The first robot of each lane is located at $(s, s - (i-1)d)$ for the Lane $i$. The next proposition states the throughput for a given time and the asymptotic throughput for this strategy.

**Proposition 4.** *Assume a circular target region with its centre at $(0,0)$ and radius $s \geq \frac{d}{2}$ and parallel lanes starting at $(s, s - (i-1)d)$ for $i \in \{1, \ldots, \lfloor \frac{2s}{d} \rfloor + 1\}$. At each Lane $i$, the first robot is located at the point $(s, s - (i-1)d)$ in the starting configuration. Then, the first robot to reach the target is located at $(s, s - (J-1)d)$, for $J = \lfloor \frac{s}{d} \rfloor + 1$, if $|s - \lfloor \frac{s}{d} \rfloor d| \leq |s - \lceil \frac{s}{d} \rceil d|$, otherwise $J = \lceil \frac{s}{d} \rceil + 1$. The throughput for a given time $T$ after the first robot reaches the target region is:*

$$f_p(T) = \frac{1}{T}\left( \sum_{i=1}^{\lfloor \frac{2s}{d} \rfloor + 1} N_i(T) \right) - \frac{1}{T}, \tag{6}$$

*for $N_i(T) = \left\lfloor \frac{vT - d_i + d_J}{d} + 1 \right\rfloor$, if $T \geq \frac{d_i - d_J}{v}$, otherwise, $N_i(T) = 0$, $d_j = s - \sqrt{s^2 - (s - (j-1)d)^2}$, and*

$$f_p = \lim_{T\to\infty} f_p(T) = \left\lfloor \frac{2s}{d} + 1 \right\rfloor \frac{v}{d}. \tag{7}$$

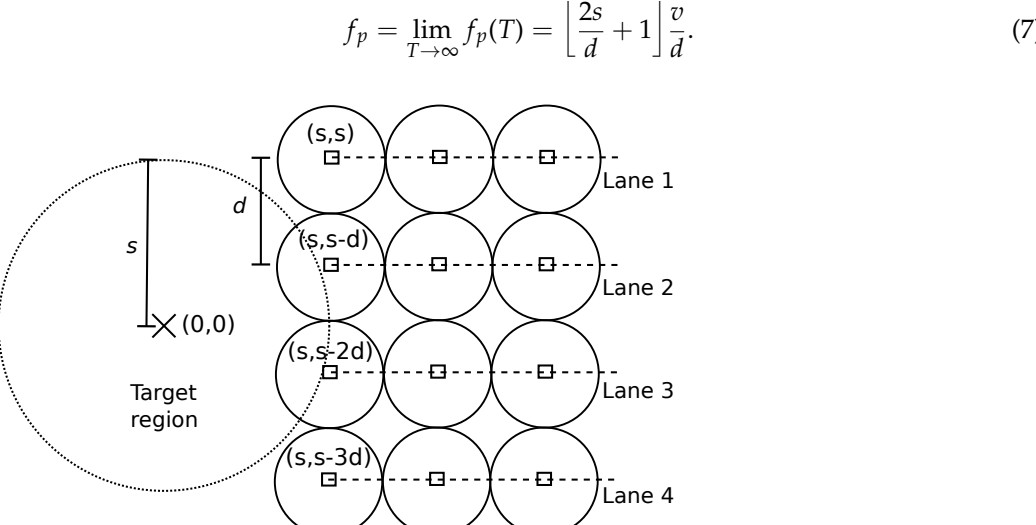

**Figure 4.** Example of the parallel lanes strategy.

### 3.3.2. Hexagonal Packing

By extending the compact lanes to more than two lanes, the robots will be packed in a hexagonal formation. An illustration of this strategy is shown in Figure 5. As it can be seen, robots from different lanes are still able to move towards the target keeping a distance $d$ from each other, even though the lanes have a distance lower than $d$.

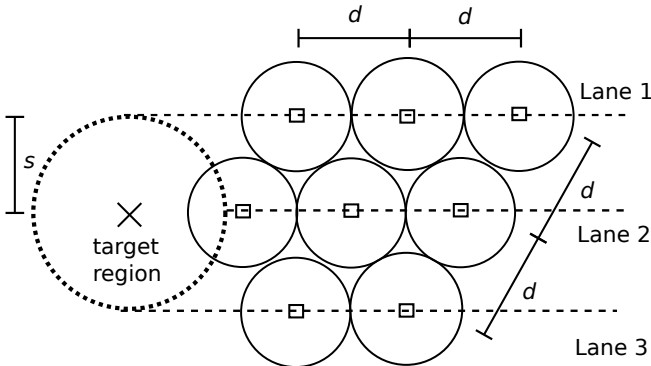

**Figure 5.** Robot lanes for hexagonal packing.

An upper bound of the asymptotic throughput for the *hexagonal packing* strategy is first computed, then the throughput for a given time using this strategy is calculated.

**Proposition 5.** *Assume robots moving at speed v, going to a circular target of radius s. The upper bound of the asymptotic throughput for the hexagonal packing strategy is*

$$f_h^{max} = \frac{2}{\sqrt{3}} \left( \frac{2s}{d} + 1 \right) \frac{v}{d}. \tag{8}$$

Proposition 5 presents an upper bound of the asymptotic throughput using hexagonal packing, but it does not tell us which is the best placement of the robots inside a corridor since the hexagonal formation can be rotated by different angles. Hence, the results about the throughput considering the placement of the hexagonal packing inside a corridor of robots going to the target region will be presented. First, however, the following definition will be needed.

**Definiton 3.** *The hexagonal packing angle $\theta$ is the angle formed by the x-axis and the line formed by any robot at position $(x, y)$ and its neighbour at $(x + d \cos(\theta), y + d \sin(\theta))$ under the target region reference frame.*

Observe that any robot at $(x, y)$ under the hexagonal packing has at most six neighbours located at $(x + d \cos(\theta), y + d \sin(\theta))$, $(x + d \cos(\theta + \frac{\pi}{3}), y + d \sin(\theta + \frac{\pi}{3}))$, ..., $(x + d \cos(\theta + \frac{5\pi}{3}), y + d \sin(\theta + \frac{5\pi}{3}))$ (Figure 6). If $\theta = \frac{\pi}{3}$, putting this value in the previous series results in the first neighbour robot being at $(x + d \cos(\pi/3), y + d \sin(\pi/3))$ and the last neighbour robot at $(x + d \cos(0), y + d \sin(0))$. This is the same result if $\theta = 0$ was used. Consequently, due to this periodicity, hexagonal packing angles in $[0, \frac{\pi}{3})$ are assumed.

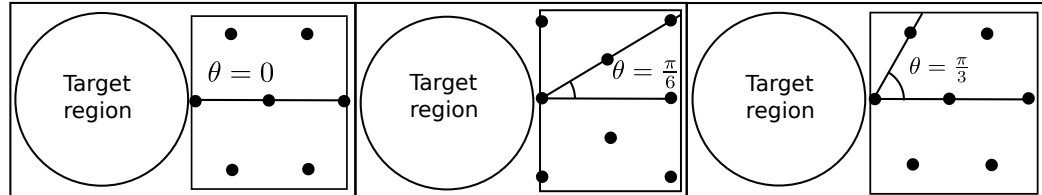

**Figure 6.** Example of hexagonal packing with different angles. The robots are the black dots.

The next proposition states the bounds of the throughput in the limit towards the infinity for hexagonal packing using an arbitrary, but fixed, hexagonal packing angle $\theta$. A fixed $\theta$ is assumed because normally in a robotic swarm the robots rely on local sensing. In order to obtain the maximum number of robots inside the corridor, all robots should know the size of the corridor and communicate by local-ranged message sending. It would take time to send information, and for all robots to adjust their orientation each time a new robot joins the swarm when using this local sensing approach.

In other words, if the corridor where the robots are going in the direction of the target is increasing over time, then $\theta$ should change over time for the optimal throughput. However, in practice, changing the hexagonal packing angle implies all robots must turn to a hexagonal packing angle $\theta^*$ depending on the size of the new rectangle based on the added robots to it to maximise the number of robots inside the corridor. In addition to the time to send messages with this parameter, more time would be needed for every robot to adapt to the updated computed $\theta^*$ because the turning speed of the robots is finite. Therefore, this paper does not handle this adjustable scenario.

**Proposition 6.** *Assume the robots using hexagonal formation coming to a circular target area with radius s such that the first robot to reach it was at time 0 at $(x_0, y_0) = (w, 0)$, for any $w \geq s$. For a given time T, the robots are going to the target at linear speed v, keeping a distance d between neighbours $(0 < d \leq 2s)$, using fixed hexagonal packing angle $\theta \in [0, \pi/3)$. The throughput for a given time is given by*

$$f_h(T, \theta) = \frac{1}{T} \sum_{x_h = -n_l^-}^{n_l^+ - 1} \left( \lfloor Y_2^R(x_h) \rfloor - \lceil Y_1^R(x_h) \rceil + 1 \right) +$$

$$\frac{1}{T} \sum_{x_h = B}^{U} \left( \lfloor Y_2^S(x_h) \rfloor - \lceil Y_1^S(x_h) \rceil + 1 \right) - \frac{1}{T}, \tag{9}$$

*for $\lfloor Y_2^R(x_h) \rfloor \geq \lceil Y_1^R(x_h) \rceil$ and $\lfloor Y_2^S(x_h) \rfloor \geq \lceil Y_1^S(x_h) \rceil$ (if for some $x_h$, either of these conditions are false, it is assumed that the respective summand for this $x_h$ is zero). Additionally, $n_l^- = \left\lfloor \frac{2s \sin(|\pi/6 - \theta|)}{\sqrt{3}d} \right\rfloor$, $n_l^+ = \left\lfloor \frac{2(vT - s) \cos(\pi/6 - \theta) + 2s \sin(|\pi/6 - \theta|)}{\sqrt{3}d} + 1 \right\rfloor$,*

$$Y_1^R(x_h) = \begin{cases} \max\left( \dfrac{\sin(\frac{\pi}{3} - \theta)x_h - \frac{s}{d}}{\cos\left(\theta - \frac{\pi}{6}\right)}, \dfrac{-\cos(\frac{\pi}{3} - \theta)x_h}{\sin\left(\frac{\pi}{6} - \theta\right)} \right), & \text{if } \theta < \pi/6, \\[12pt] \max\left( \dfrac{\sin(\frac{\pi}{3} - \theta)x_h - \frac{s}{d}}{\cos\left(\theta - \frac{\pi}{6}\right)}, \dfrac{\frac{vT-s}{d} - \cos(\frac{\pi}{3} - \theta)x_h}{\sin\left(\frac{\pi}{6} - \theta\right)} \right), & \text{if } \theta > \pi/6, \\[12pt] \dfrac{x_h}{2} - \dfrac{s}{d}, & \text{if } \theta = \pi/6, \end{cases}$$

$$Y_2^R(x_h) = \begin{cases} \min\left( \dfrac{\sin(\frac{\pi}{3} - \theta)x_h + \frac{s}{d}}{\cos\left(\theta - \frac{\pi}{6}\right)}, \dfrac{\frac{vT-s}{d} - \cos(\frac{\pi}{3} - \theta)x_h}{\sin\left(\frac{\pi}{6} - \theta\right)} \right), & \text{if } \theta < \pi/6, \\[12pt] \min\left( \dfrac{\sin(\frac{\pi}{3} - \theta)x_h + \frac{s}{d}}{\cos\left(\theta - \frac{\pi}{6}\right)}, \dfrac{-\cos(\frac{\pi}{3} - \theta)x_h}{\sin\left(\frac{\pi}{6} - \theta\right)} \right), & \text{if } \theta > \pi/6, \\[12pt] \dfrac{x_h}{2} + \dfrac{s}{d}, & \text{if } \theta = \pi/6, \end{cases}$$

$$B = \begin{cases} \left\lceil \dfrac{2(\sin(\pi/3 - \theta)(c_x - l_x) + \cos(\pi/3 - \theta)(y_0 - l_y - s))}{\sqrt{3}d} \right\rceil, & \text{if } T > \dfrac{s}{v}, \\[12pt] \left\lceil -\dfrac{2\sqrt{2svT - (vT)^2}}{\sqrt{3}d} \sin\left(\theta + \dfrac{\pi}{6}\right) \right\rceil, & \text{otherwise,} \end{cases}$$

for $c_x = x_0 + vT - s$ and $(l_x, l_y) = \mathrm{argmin}_{(x,y) \in Z} |vT - s + x_0 - x| + |y_0 - y|$, if $T > \frac{s}{v}$, otherwise, $(l_x, l_y) = (x_0, y_0)$, where $Z$ is the set of robot positions inside the rectangle measuring $vT - s \times 2s$ for $vT - s > 0$. If $T > \frac{s}{v}$ or $\arctan\left( \frac{\frac{s}{2} - \sin(\theta)(vT - s)}{\frac{\sqrt{3}s}{2} + \cos(\theta)(vT - s)} \right) < \frac{\pi}{2} - \theta$, $U = \left\lfloor \frac{2(\sin(\pi/3 - \theta)(c_x - l_x) + \cos(\pi/3 - \theta)(y_0 - l_y) + s)}{\sqrt{3}d} \right\rfloor$, otherwise, $U = \left\lfloor \frac{2\sqrt{2svT - (vT)^2}}{\sqrt{3}d} \cos\left(\theta - \frac{\pi}{3}\right) \right\rfloor$. In addition, $Y_1^S(x_h) = \frac{dx_h - C_{-\theta,x} + \sqrt{3}C_{-\theta,y} - \sqrt{\Delta(x_h)}}{2d}$ and

$$Y_2^S(x_h) = \begin{cases} \min(L(x_h), C_2(x_h)) - 1, & \text{if } \min(L(x_h), C_2(x_h)) \\ & \quad = \lfloor L(x_h) \rfloor \text{ and } T > \dfrac{s}{v}, \\[6pt] \min(L(x_h), C_2(x_h)), & \text{otherwise,} \end{cases} \tag{10}$$

$C_{-\theta} = \begin{bmatrix} \cos(-\theta) & -\sin(-\theta) \\ \sin(-\theta) & \cos(-\theta) \end{bmatrix} \begin{bmatrix} c_x - l_x \\ y_0 - l_y \end{bmatrix}$, $\Delta(x_h) = 4s^2 - \left( \sqrt{3}(dx_h - C_{-\theta,x}) - C_{-\theta,y} \right)^2$,

$C_2(x_h) = \frac{dx_h - C_{-\theta,x} + \sqrt{3}C_{-\theta,y} + \sqrt{\Delta(x_h)}}{2d}$, $L(x_h) = \frac{\sin(\frac{\pi}{2} - \theta)(dx_h - C_{-\theta,x}) + \cos(\frac{\pi}{2} - \theta)C_{-\theta,y}}{d\sin(\frac{5\pi}{6} - \theta)}$, if $T > \frac{s}{v}$, otherwise $L(x_h) = \frac{\sin(\frac{\pi}{2} - \theta)x_h}{\sin(\frac{5\pi}{6} - \theta)}$, and

$$\lim_{T \to \infty} f_h(T, \theta) \in \left( \frac{4vs}{\sqrt{3}d^2} - \frac{2v\cos(\theta - \pi/6)}{\sqrt{3}d}, \frac{4vs}{\sqrt{3}d^2} + \frac{2v\cos(\theta - \pi/6)}{\sqrt{3}d} \right]. \tag{11}$$

The upper and lower bounds presented on (11) are below or equal the maximum asymptotic throughput presented by the Proposition 5, Equation (8). The result of the Proposition 5 only concerns the maximum asymptotic throughput and does not consider the hexagonal packing angle $\theta$, while Proposition 6 gives a lower bound and tightens the bounds for a given $\theta$. Figure 7 presents an example comparison of these equations for two different values of $s$. As expected, the maximum asymptotic throughput under the optimal density assumption (in (8)) is a possible value of the throughput using hexagonal packing and is above or equal to the interval in (11) for any given $\theta$. However, for practical robotic swarms applications, a certain hexagonal packing angle must be fixed depending on the expected height of the corridor, target size and the minimum distance between the robots, resulting in a throughput below or equal to the upper value presented in Proposition 5.

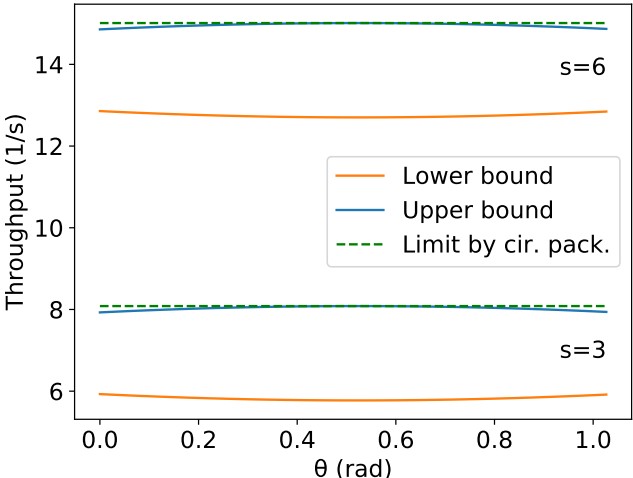

**Figure 7.** Limit given by (8) using the circle packing results and the lower and upper bounds of the hexagonal packing limit by (11) for $\theta \in [0, \pi/3)$, $d = 1$ m, $v = 1$ m/s and $s \in \{3, 6\}$ m.

On the other hand, due to the discontinuities of (9), it is difficult to obtain an exact $\theta$ that maximises the throughput given the other parameters. In addition, there is no specific value of $\theta$ that achieves the maximum throughput for all possible values of the other parameters. Interestingly, given a fixed sub-interval of $\theta$, depending on the number of sample values, new local maxima and minima can arise from these discontinuities. Additionally, a different parity of the number of samples can produce a global maximum in even or odd interval points. To illustrate this, Figures 8–11 present the result of this equation for some randomly generated parameters and a different number of samples of $\theta$ equally spaced and taken from the domain interval, that is, from 0 to $\pi/3$, including these values. Two different orders of magnitude are chosen for the number of equally spaced points in each plot (a small one, about two orders, and a large one of seven orders), and different parities are also given (99 and 100 for the small order, and $10^7$ and $10^7 + 1$ for the large one).

In Figures 8–11, $\theta$ is over the $x$-axis, and the number of robots inside the given rectangle is over the $y$-axis. These plots use $v = 1$ m/s. The maximum value in each image is represented by an orange circle, and a rectangle represents the maximum between the left and the right image. No square means the maximum values in both sides are equal. Each one of the Figures 8–11 presents two different sets of parameters. In Figures 8 and 9, 99 equally spaced values are shown for $\theta \in [0, \pi/3)$ on the left-hand side images and 100 on the right-hand side; then, the maximum on each side is compared, and the best one is chosen. The same is performed in Figures 10 and 11, but using $10^7$ and $10^7 + 1$. Figures 8a, 9a, 10b and 11b show an example that $\theta \approx \pi/6$ reaches the maximum throughput, and in Figures 8c,d and 10c,d, the maximum is at $\theta = 0$. Moreover, Figure 9c,d have their maximum for $\theta$ different from the other examples. Figure 8c,d have the same maximum, despite the plots being different. This also occurs in Figures 10c,d and 11c,d. If the parameters are known, one can find an approximate best candidate for $\theta$ by searching several values, as presented. However, as far as the authors know, obtaining the true value which maximises that equation by a closed-form is an open problem.

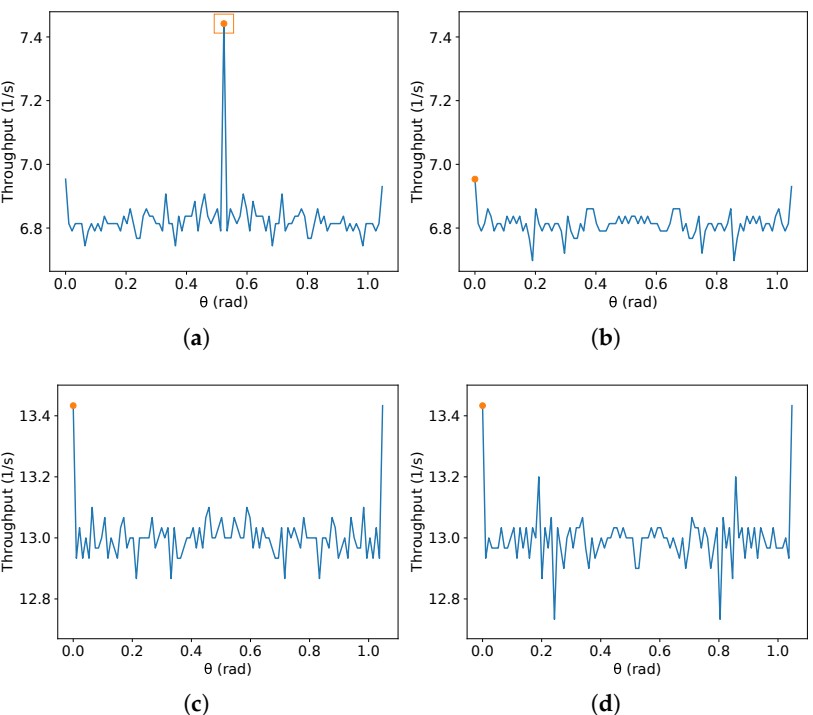

**Figure 8.** Examples of (9) varying $\theta$ from 0 to $\frac{\pi}{3}$ for different and randomly generated values of $T$, $s$, and $d$. It continues in Figure 9. (**a**) For 99 samples, $T = 43$ s, $s = 3$ m, $d = 1$ m. (**b**) For 100 samples, $T = 43$ s, $s = 3$ m, $d = 1$ m. (**c**) For 99 samples, $T = 30$ s, $s = 2.5$ m and $d = 0.66$ m. (**d**) For 100 samples, $T = 30$ s, $s = 2.5$ m and $d = 0.66$ m.

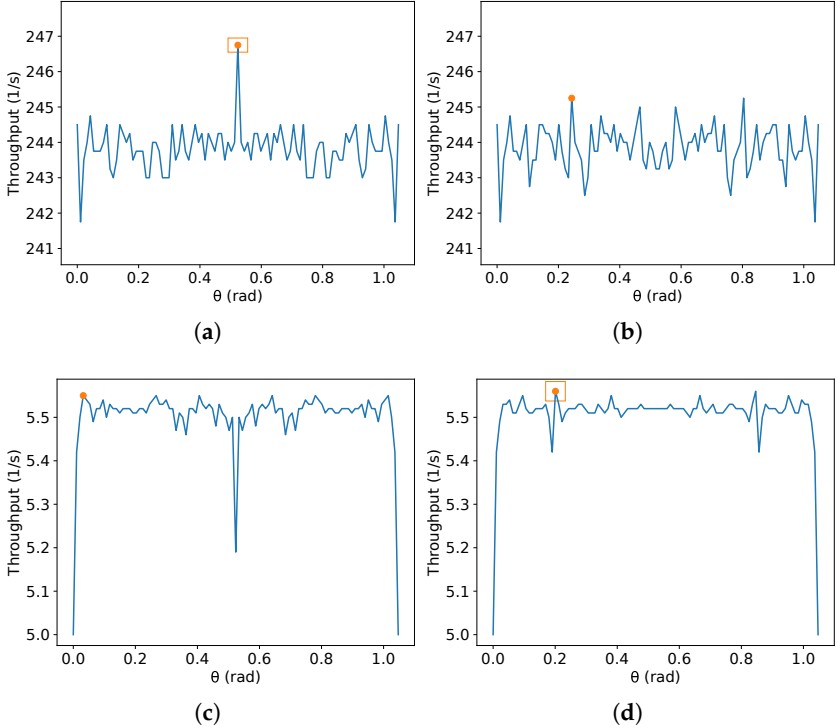

**Figure 9.** Continuation of Figure 8: examples of (9) varying $\theta$ from 0 to $\frac{\pi}{3}$ for different and randomly generated values of $T$, $s$ and $d$. (**a**) For 99 samples, $T = 4$ s, $s = 2$ m and $d = 0.13$ m. (**b**) For 100 samples, $T = 4$ s, $s = 2$ m and $d = 0.13$ m. (**c**) For 99 samples, $T = 100$ s, $s = 2.40513$ m and $d = 1$ m. (**d**) For 100 samples, $T = 100$ s, $s = 2.40513$ m and $d = 1$ m.

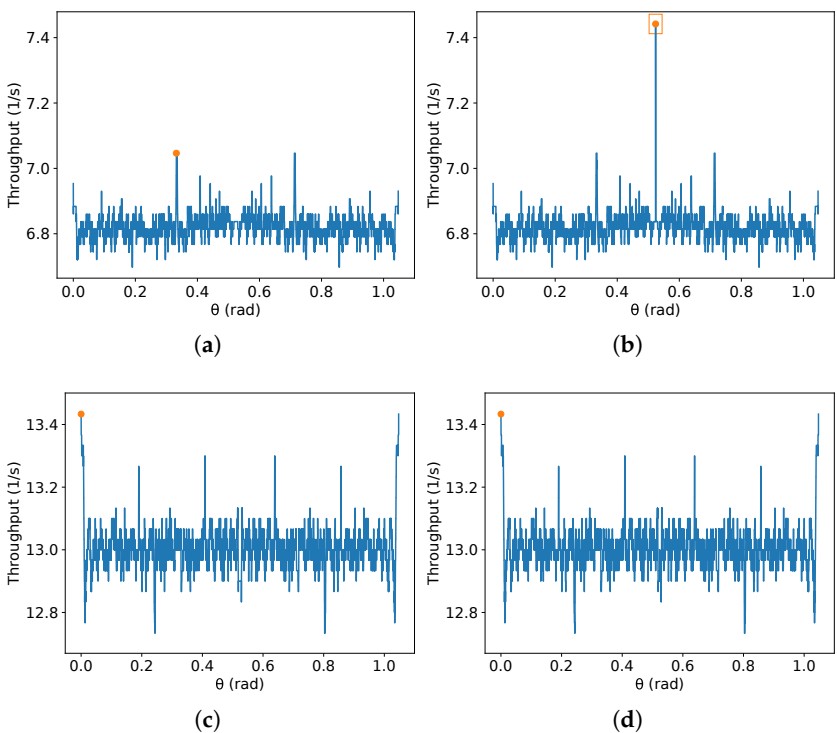

**Figure 10.** Similar to Figures 8 and 9 but using $10^7$ and $10^7 + 1$ equally spaced points for $\theta \in [0, \pi/3)$. It continues in Figure 11. (**a**) For $10^7$ samples, $T = 43$ s, $s = 3$ m, $d = 1$ m. (**b**) For $10^7 + 1$ samples, $T = 43$ s, $s = 3$ m, $d = 1$ m. (**c**) For $10^7$ samples, $T = 30$ s, $s = 2.5$ m and $d = 0.66$ m. (**d**) For $10^7 + 1$ samples, $T = 30$ s, $s = 2.5$ m and $d = 0.66$ m.

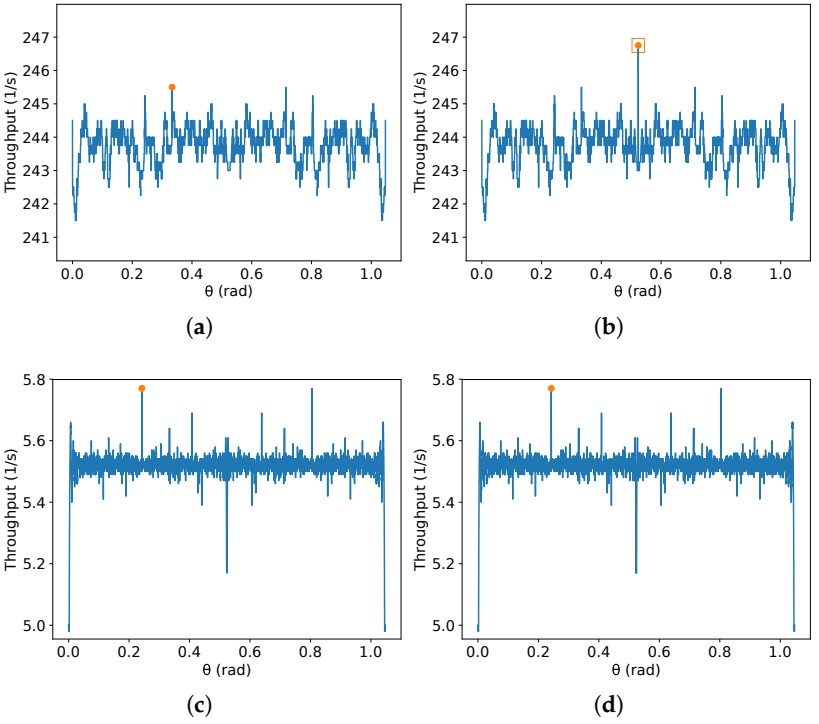

**Figure 11.** Continuation of Figure 10: examples similar to Figures 8 and 9 but using $10^7$ and $10^7 + 1$ equally spaced points for $\theta \in [0, \pi/3)$. (**a**) For $10^7$ samples, $T = 4$ s, $s = 2$ m and $d = 0.13$ m. (**b**) For $10^7 + 1$ samples, $T = 4$ s, $s = 2$ m and $d = 0.13$ m. (**c**) For $10^7$ samples, $T = 100$ s, $s = 2.40513$ m and $d = 1$ m. (**d**) For $10^7 + 1$ samples, $T = 100$ s, $s = 2.40513$ m and $d = 1$ m.

Additionally, notice that whenever the number of samples is odd, the value $\theta = \pi/6$ is sampled. Observe in these figures that when the maximum is at $\theta = \pi/6$, it tends to be higher than the maximum found without considering it. For instance, compare the maximum found on the pairs (a) and (b) in Figures 8–11. On the other hand, $\theta = \pi/6$ is not always the optimal value. Thus, the authors suggest to compute first the value for $\theta = \pi/6$, then compare it with the result for a search for the maximum for any chosen number of samples in the interval from $\theta \in [0, \pi/3)$.

### 3.3.3. Touch and Run Strategy

Now, the *touch and run* strategy is discussed. Since a robot should spend as little time as possible near the target, a simple scenario is imagined where robots travel in predefined curved lanes and tangent to the target area where they spend minimum time on the target. To avoid collisions with other robots, the trajectory of a robot nearby the target is circular, and the distance between each robot must be at least $d$ at any part of the trajectory. Hence, no lane crosses another, and each lane occupies a region defined by an angle in the target area, denoted by $\alpha$ and shown in Figure 12a.

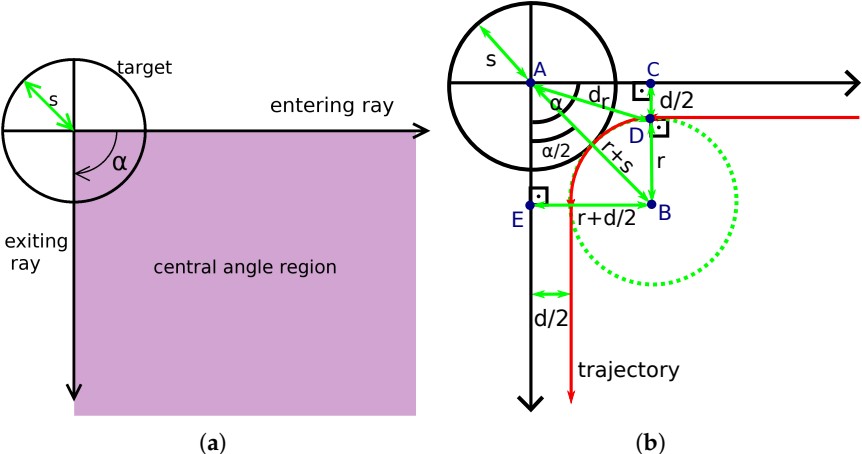

(a)                                        (b)

**Figure 12.** Illustration of the touch and run strategy. (**a**) Central angle region and its exiting and entering rays defined by the angle $\alpha$. (**b**) Trajectory of a robot next to the target in red.

Figure 12b shows the trajectory of a robot towards the target region following that strategy. This figure also shows the relationship between the target area radius ($s$), the minimum safety distance between the robots ($d$), the turning radius ($r$), the central region angle ($\alpha$) and the distance from the target centre for a robot to begin turning ($d_r$)—used as justification for (12) and (13). The green dashed circle represents the whole turning circle. The robot first follows the boundary of the central angle region—that is, the entering ray—at a distance of $d/2$. Then, it arrives at a distance of $s$ of the target centre using a circular trajectory with a turning radius $r$. Due to the trajectory being tangent to the target shape, it is close enough to consider that the robot reached the target region.

Finally, the robot leaves the target by following the second boundary of the central angle region—that is, the exiting ray—at a distance of $d/2$. Depending on the value of $\alpha$, it is possible to fit several of these lanes around the target. For example, in Figure 13, when $\alpha = \pi/2$, it is possible to fit four lanes. In this figure, robots are black dots, and $d_o$ is the desired distance between the robots in the same lane—which is calculated depending on the values of $d$, $s$, $r$ and the number of lanes $K$ as shown later. When robots of all lanes simultaneously occupy the target region, their positions are the vertices of a regular polygon—it is represented in the figure by a grey square inside the target region.

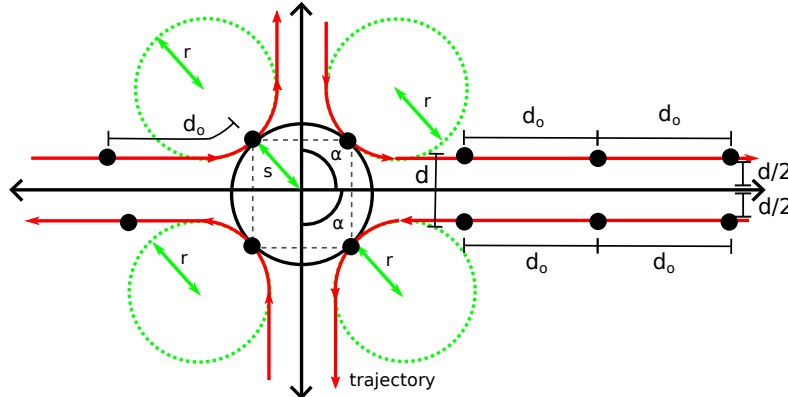

**Figure 13.** Theoretical trajectory in red, for $\alpha = \pi/2$ and $K = 4$.

The lemma below concerns the distance to the target centre where the robots start turning on the curved path. It will also be useful in the discussion about experiments using this strategy in Section 4.4.

**Lemma 2.** *The distance $d_r$ to the target centre for the robot to start turning is*

$$d_r = \sqrt{s(2r + s) - rd}. \tag{12}$$

Now, a lemma about the turning radius is presented, and then the domain of $K$ and $\alpha$ are defined in order to calculate the throughput for the touch and run strategy.

**Lemma 3.** *The central region angle $\alpha$, the minimum distance between the robots $d$ and the turning radius $r$ are related by*

$$r = \frac{s \sin(\alpha/2) - d/2}{1 - \sin(\alpha/2)}. \tag{13}$$

**Proposition 7.** *Let $K$ be the number of curved trajectories around the target area, $\alpha$ be the angle of each central area region, and $r$ the turning radius of the robot for the curved trajectory of this central area region. For a given $d > 0$ and $s \geq d/2$, the domain of $K$ is*

$$3 \leq K \leq \frac{\pi}{\arcsin\left(\frac{d}{2s}\right)}, \text{ and} \tag{14}$$

$$\alpha = \frac{2\pi}{K}. \tag{15}$$

Now that the correct parametrisation has been determined for the touch and run strategy, its throughput is obtained in the next proposition.

**Proposition 8.** *Assuming the touch and run strategy and that the first robot of every lane begins at the same distance from the target, given a target radius $s$, the constant linear robot speed $v$, a minimum distance between robots $d$, and the number of lanes $K$, the throughput for a given instant $T$ is calculated by*

$$f_t(K, T) = \frac{1}{T}\left(K\left\lfloor \frac{vT}{d_o} + 1 \right\rfloor - 1\right), \text{ for} \tag{16}$$

$$d_o = \max(d, d'), \text{ and} \tag{17}$$

$$d' = \begin{cases} r(\pi - \alpha) + \frac{d - 2r\cos(\alpha/2)}{\sin(\alpha/2)}, & \text{if } 2r\cos(\alpha/2) < d, \\ 2r\arcsin\left(\frac{d}{2r}\right), & \text{otherwise}, \end{cases} \tag{18}$$

with *r* obtained from (13). In addition,

$$f_t(K) = \lim_{T \to \infty} f_t(K, T) = \frac{Kv}{d_o}. \tag{19}$$

Figure 14 presents examples of (19) for some parameters. Observe that the maximum throughput for different values of $s, d$ and $v$ can be found by a linear search in the interval obtained by (14).

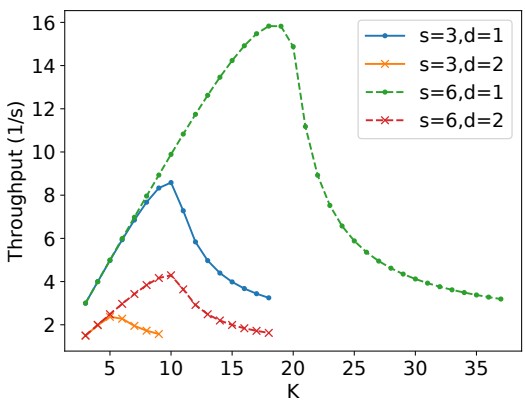

**Figure 14.** Plot of the asymptotic throughput of the touch and run strategy (given by (19)) for some values of *s* and *d*, in metres, and $v = 1\,\text{m/s}$, for the interval of values for *K* obtained by (14).

### 3.3.4. Comparison of the Strategies

The parallel lanes strategy has the lowest of the limits concerning $u = \frac{s}{d}$, the ratio between the radius of the target region and the minimum distance between the robots. However, its asymptotic value is still higher than the minimum possible asymptotic throughput for hexagonal packing just for some values of *u*. This section will make explicit the dependence on the argument *u* in every throughput function defined previously to compare them to this ratio. Let $f_p(u) = \lim_{T \to \infty} f_p(T, u)$ and $f_h^{min}(u)$ be the asymptotic throughput for the parallel lanes strategy and the lower asymptotic throughput for the hexagonal packing strategy for a ratio *u*, respectively. Hence, by Proposition 4, $f_p(u) = \lfloor 2u + 1 \rfloor \frac{v}{d}$, and by (11) using $\theta = \pi/6$ as it minimises the lower bound of $\lim_{T \to \infty} f(T, \theta)$ in Proposition 6, $f_h^{min}(u) = \frac{2}{\sqrt{3}}(2u - 1)\frac{v}{d}$.

**Proposition 9.** *There are some* $u < \frac{\sqrt{3}+2}{4-2\sqrt{3}}$ *such that* $f_p(u) > f_h^{min}(u)$, *and for every* $u \geq \frac{\sqrt{3}+2}{4-2\sqrt{3}}$, $f_p(u) \leq f_h^{min}(u)$.

Figure 15 shows an example of $f_h^{min}(u)$, $f_p(u)$ and the maximum possible asymptotic throughput of the hexagonal packing $f_h^{max}(u) = \frac{2}{\sqrt{3}}(2u + 1)\frac{v}{d}$ for $u \in [0, 10]$. Observe that, from the left side of $u = 7$, $f_p(u)$ has some values above $f_h^{min}(u)$ even though they are below $f_h^{max}(u)$ for every *u*.

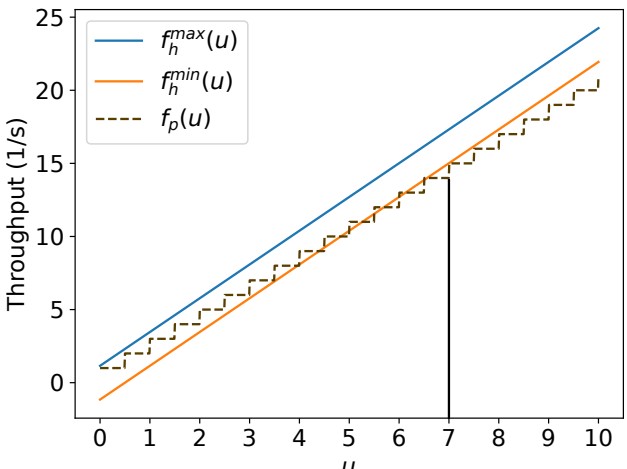

**Figure 15.** Example of $u$ values such that $f_h^{max}(u) > f_p(u)$ for $v = 1$ m/s and $d = 1$ m.

Because of this proposition, for values of $u \geq \frac{\sqrt{3}+2}{4-2\sqrt{3}} \approx 7$, the hexagonal packing strategy at the limit will have higher throughput than parallel lanes. However, for values $u < \frac{\sqrt{3}+2}{4-2\sqrt{3}}$, there is the possibility of the parallel lanes strategy being better than hexagonal packing. As there is not an exact asymptotic throughput for the hexagonal packing strategy for a given angle $\theta$, one can numerically find the best $\theta$ using large values of $T$ on (9); then, after choosing $\theta$, the numerical approximation of the asymptotic throughput using this fixed $\theta$ and those $T$ values is calculated. This result can be compared with the throughput for the same large values of $T$ for the parallel lanes strategy using (6). Furthermore, in a scenario with the target region only being accessed by a corridor with a finite height, the maximum time $T$ can be inferred by its size, and then the exact throughput for this specific value can be calculated by (9) and (6) as stated before, but using only this specific value $T$, instead of a set of large values, to decide which strategy is more suitable.

Let $f_h(T, \theta, u)$ and $f_p(T, u)$ be (9) and (6) making explicit the parameter $u$. Let $\theta^*$ be the outcome from the search of the $\theta$, which maximises $f_h(T, \theta, u)$ by numeric approximation. Thus, define $f_h(T, u) = f_h(T, \theta^*, u)$. Figure 16 illustrates the result of the procedure mentioned above for $T = 10{,}000$ for 100 equally spaced values of $u \in [0, 7]$ and seeking the maximum throughput using 1000 evenly spaced points between $[0, \pi/3)$ to find the best $\theta$ for the hexagonal packing strategy. Then, it is compared with the result for $\theta = \pi/6$ as explained previously when Figures 8–11 were discussed. Observe that for $u \in [0.5, 0.9]$ there is some values for which $f_h(10{,}000, u) < f_p(10{,}000, u)$. Figure 17 shows this by 100 equally spaced values of $u \in [0.4, 1]$ for different values of $v$. This occurs because, for such values of $u$, using square packing fits more robots inside the circle over the time than hexagonal packing, as shown in Section 4.5.

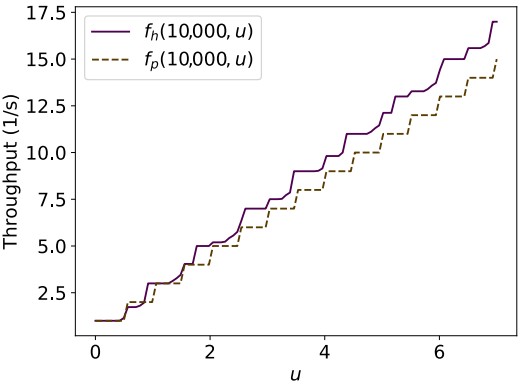

**Figure 16.** Comparison of $f_p(T, u)$ and $f_h(T, u)$ for $u \in [0, 7]$, $T = 10{,}000$ s, $v = 1$ m/s and $d = 1$ m.

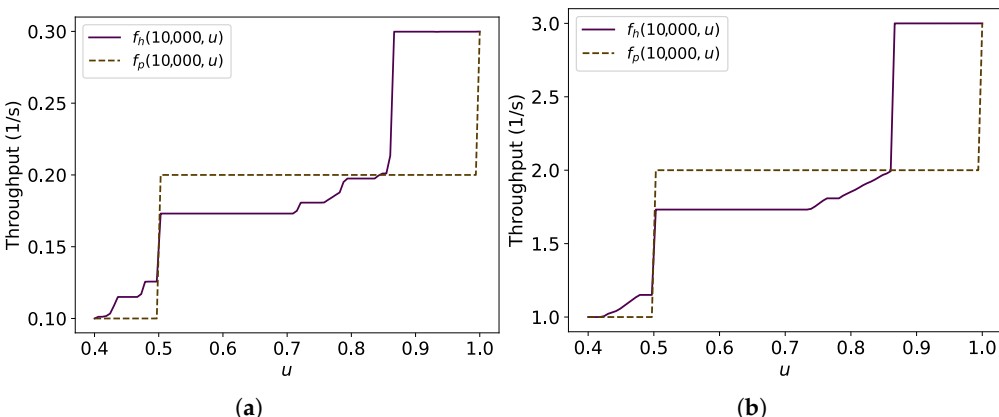

**Figure 17.** Comparison of $f_p$ and $f_h$ for $u \in [0.4,1]$, $T = 10{,}000$ s, $v \in \{0.1,1\}$ m/s and $d = 1$ m. The difference in the lines of $f_h$ is due to $\theta^*$ being different for each value of $v$. (**a**) $v = 0.1$ m/s; (**b**) $v = 1$ m/s.

Additionally, the asymptotic throughput of the touch and run strategy, $f_t(u) = \lim_{T \to \infty} f_t(T, u)$, for higher values of $u$ is greater than the maximum possible asymptotic value of the hexagonal packing $f_h^{max}(u) = \frac{2}{\sqrt{3}}(2u + 1)\frac{v}{d}$, as shown later by numeric experimentation. Before presenting this result, it is necessary to verify which values of $u$ are allowed by $f_t(u)$ and to express the asymptotic throughput of the touch and run strategy from Proposition 8 in terms of the ratio $u$.

From Proposition 7, the possible number of lanes $K$ is in $\{3, \ldots, K(u)\}$ with $K(u) = \left\lfloor \frac{\pi}{\arcsin\left(\frac{1}{2u}\right)} \right\rfloor$. Consequently, $f_t(u)$ is only allowed for any $u \geq \frac{1}{\sqrt{3}}$. In fact, by Proposition 7, $K \geq 3$, then $\frac{\pi}{\arcsin\left(\frac{1}{2u}\right)} \geq \left\lfloor \frac{\pi}{\arcsin\left(\frac{1}{2u}\right)} \right\rfloor \geq 3 \Rightarrow \frac{\pi}{3} \geq \arcsin\left(\frac{1}{2u}\right) \Leftrightarrow \sin\left(\frac{\pi}{3}\right) \geq \frac{1}{2u} \Leftrightarrow \frac{\sqrt{3}}{2} \geq \frac{1}{2u} \Leftrightarrow u \geq \frac{1}{\sqrt{3}}$.

The algebraic manipulations for expressing the asymptotic throughput of the touch and run strategy from Proposition 8 is shown below in terms of the ratio $u$. The asymptotic throughput expressed in (19) is

$$\frac{Kv}{d_o} = \frac{K}{\frac{d_o}{d}}\frac{v}{d} = \frac{K}{\frac{\max(d,d')}{d}}\frac{v}{d} = \frac{K}{\max(1, \frac{d'}{d})}\frac{v}{d'} \tag{20}$$

for an integer $K \in \{3, \ldots, K(u)\}$. From (15), $\alpha = \frac{2\pi}{K}$, and, from (13), $\frac{r}{d} = \frac{\frac{s}{d}\sin(\alpha/2) - \frac{d}{2d}}{1 - \sin(\alpha/2)} = \frac{u\sin(\frac{\pi}{K}) - \frac{1}{2}}{1 - \sin(\frac{\pi}{K})} \stackrel{\text{def}}{=} r(u, K)$, resulting in

$$
\begin{aligned}
\frac{d'}{d} &= \begin{cases} \frac{r}{d}(\pi - \alpha) + \frac{d - 2r\cos(\alpha/2)}{d\sin(\alpha/2)}, & \text{if } 2r\cos(\alpha/2) < d, \\ 2\frac{r}{d}\arcsin\left(\frac{d}{2r}\right), & \text{otherwise,} \end{cases} \quad \text{[by (18)]} \\
&= \begin{cases} \frac{r}{d}\left(\pi - \frac{2\pi}{K}\right) + \frac{1 - 2\frac{r}{d}\cos(\frac{\pi}{K})}{\sin(\frac{\pi}{K})}, & \text{if } 2\frac{r}{d}\cos(\frac{\pi}{K}) < 1, \\ 2\frac{r}{d}\arcsin\left(\left(2\frac{r}{d}\right)^{-1}\right), & \text{otherwise,} \end{cases} \\
&= \begin{cases} r(u,K)\left(\pi - \frac{2\pi}{K}\right) + \frac{1 - 2r(u,K)\cos(\frac{\pi}{K})}{\sin(\frac{\pi}{K})}, \\ \qquad\qquad \text{if } 2r(u,K)\cos(\frac{\pi}{K}) < 1, \\ 2r(u,K)\arcsin\left(\frac{1}{2r(u,K)}\right), \quad \text{otherwise,} \end{cases} \\
&\stackrel{\text{def}}{=} d'(u, K).
\end{aligned} \tag{21}
$$

Thus, from (20) and (21), $f_t(u, K) = \frac{K}{\max(1, d'(u,K))}\frac{v}{d}$, and the upper throughput for the touch and run strategy in terms of $u$ is given by $f_t(u) = \max_{K \in \{3, \ldots, K(u)\}} f_t(u, K) =$

$\max_{K\in\{3,\dots,K(u)\}} \frac{K}{\max(1,d'(u,K))} \frac{v}{d} = \frac{K^*(u)}{\max(1,d'(u,K^*(u)))} \frac{v}{d}$, for some function $K^*(u)$ that finds this maximum in $\{3,\dots,K(u)\}$. Similarly, for a fixed maximum time $T$, by (16), $f_t(T,u) = \max_{K\in\{3,\dots,K(u)\}} f_t(K,T,u)$.

Figure 18 presents a comparison of the asymptotic throughput $f_t(u)$ and the lower and upper values of the asymptotic throughput of the hexagonal packing $f_h^{min}(u)$ and $f_h^{max}(u)$ for values of $u$ ranging from $1/\sqrt{3}$ to 1000. Observe that the asymptotic throughput of the touch and run strategy is greater than the maximum possible asymptotic throughput of the hexagonal packing strategy for almost all values of $u$, except for some in $(1.12, 1.25)$ (Figure 18b).

Additionally, numerical experiments for $f_t(T,u)$ and $f_h(T,u)$ are performed using fixed time $T = 10{,}000$ in (16), (9) and $u \in [1/\sqrt{3}, 7]$. For finding $\theta^*$, the same procedure is applied, which was described before to compare $f_h(T,u)$ and $f_p(T,u)$. Figure 19 shows the result. It suggests the touch and run strategy has higher throughput than hexagonal packing for large values of $T$. Although hexagonal packing has lower asymptotic throughput than the touch and run strategy for almost all $u$ values, it is suitable for $u < \frac{1}{\sqrt{3}}$ whenever it surpasses the parallel lanes strategy.

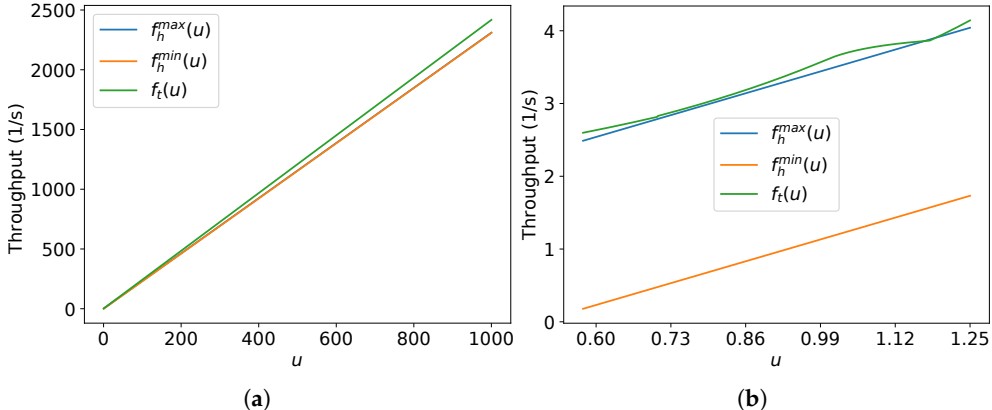

**Figure 18.** Graph varying $u$ for $f_h^{min}(u)$, $f_h^{max}(u)$ and $f_t(u)$ with $v = 1$ m/s and $d = 1$ m for different intervals of $u$. In (**a**), $f_h^{min}(u)$ and $f_h^{max}(u)$ are almost overlapped. In (**b**), $f_t(u) > f_h^{max}(u)$ for all $u$, except in an interval within $(1.12,1.25)$. (**a**) $u \in [1/\sqrt{3}, 1000]$; (**b**) $u \in [1/\sqrt{3}, 1.25]$.

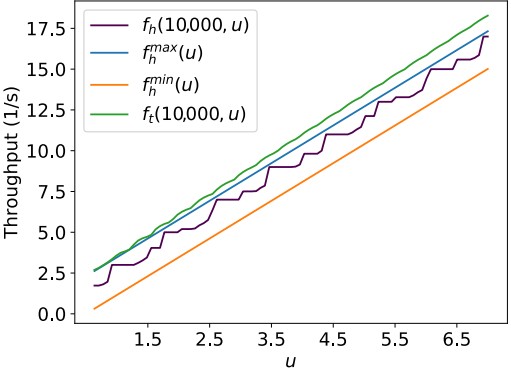

**Figure 19.** Example for $T = 10{,}000$ s, $v = 1$ m/s, $d = 1$ m and 100 equally spaced points of $u \in [1/\sqrt{3}, 7]$. $f_h(T,u) < f_t(T,u)$, albeit $f_h^{max}(u) \geq f_t(T,u)$ for a few values of $u < 1.5$.

For real-world applications and assuming the robots are constantly at maximum linear speed and at fixed distance between other robots, the hexagonal packing strategy is adequate for a situation where the target is placed in a constrained region, for example, walls in north and south positions. In this example, the number of lanes used in the touch and run strategy would be reduced because of the surrounding walls. In an unconstrained

scenario, if the ratio $u$ and the maximum time $T$ are known, the throughput value of the hexagonal packing strategy from (9) (for the $\theta$ which maximises it) can be compared with the throughput of the touch and run strategy from (16) (for $K^*(u)$) to choose which strategy should be applied. However, assuming a constant speed and a fixed minimum distance between robots in a swarm is not practical because other robots influence the movement in the environment. Hence, these strategies are the inspiration to propose novel algorithms based on potential fields for robotic swarms in [38].

## 4. Experiments and Results

To evaluate this approach, several simulations were executed using the Stage robot simulator [37] for testing the equations presented in the theoretical section (Section 3). Hyperlinks to the video of executions are available in the captions of each corresponding figure. They are in real-time so that the reader can compare the time and screenshots presented in the figures in this section with those in the supplied videos (The source codes of each experimented strategy are in https://github.com/yuri-tavares/swarm-strategies, accessed on 12 June 2022).

Experiments were executed for all strategies considering $s > 0$. We could not make experiments for point-like targets because a point with a fixed value is nearly impossible to be reached by a moving robot in Stage computer simulations due to the necessity of exact synchronization of the sampling frequency of positions made by the simulator and the speed of the robot. Hence, a circular area with a radius $s > 0$ around the target must be used to identify that a robot reached it. After presenting the experiments and results for all strategies for circular target region with radius $s > 0$, they are compared experimentally considering the analysis previously discussed in Section 3.3.4.

It is saved for each robot its arrival time in milliseconds since the start of the experiment. The arrival time of every robot is subtracted by the arrival time of the first robot. By doing so, the experiment is assumed to begin in time $T = 0$ without worrying about the initial inertia. After this, the number of robots ($N$) is registered for each time value ($T$).

To alleviate some of the numerical errors caused by the floating-point representation, rounding on the 13th decimal place was used before using floor and ceiling functions on the equations presented. For example, in contemporary computers, by using double variables in C or float in Python, if you divide 9.6 by 1.6, the result is 5.999999999999999 for 15 decimal places formatting, but it should be 6. If the floor function was applied to the previous result, the outcome would be 5 instead of the expected 6.

For all experiments in this section, the robots are distant from each other by $d = 1$ m. In the figures of this section, black robots indicate they reached the target, and red did not. In addition, the experiments shown on this section were not repeated because the linear speed and initial positions are constant, so there is no random aspect, and the same results are obtained for different runs.

### 4.1. Compact Lanes

For compact lanes simulations, $v = 1$ m/s, and the first robot to reach the target is at the bottom lane and starts at the target. For a target area radius $s$, such that $0 < s < \sqrt{3}d/4$, $s = 0.3$ m, and for $\sqrt{3}d/4 \leq s < d/2$, $s = 0.45$ m. Figure 20 shows screenshots of the simulation using $s = 0.3$ m during $T = 7.1$ s and Figure 21 for $s = 0.45$ m and $T = 10.1$ s.

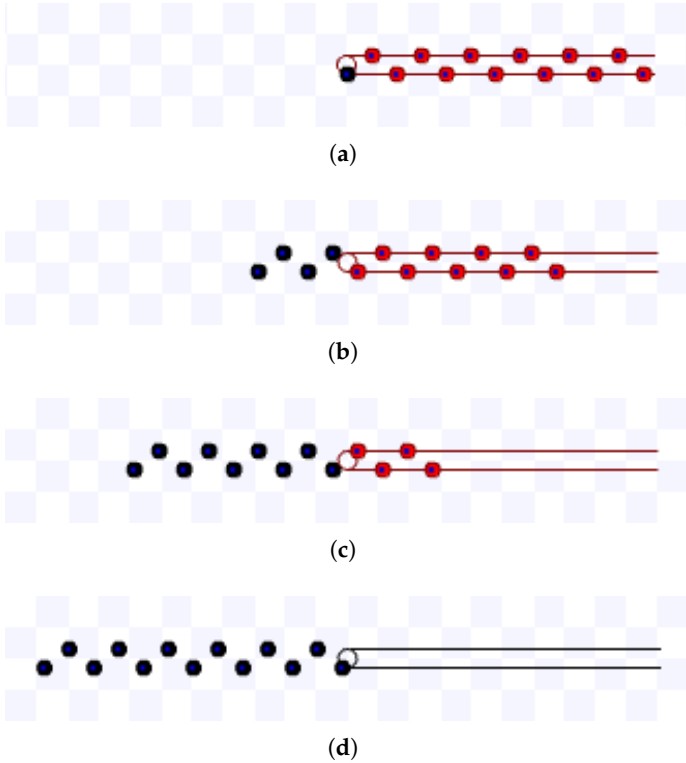

**Figure 20.** Simulation on Stage for compact lanes strategy using $s = 0.3$ m, $d = 1$ m during $T = 7.1$ s. Available on https://youtu.be/e1cWJzWhQmQ, accessed on 12 June 2022. (**a**) 0 s: beginning of the simulation; (**b**) After 2.7 s; (**c**) After 6.7 s; (**d**) 5 s: ending of the simulation.

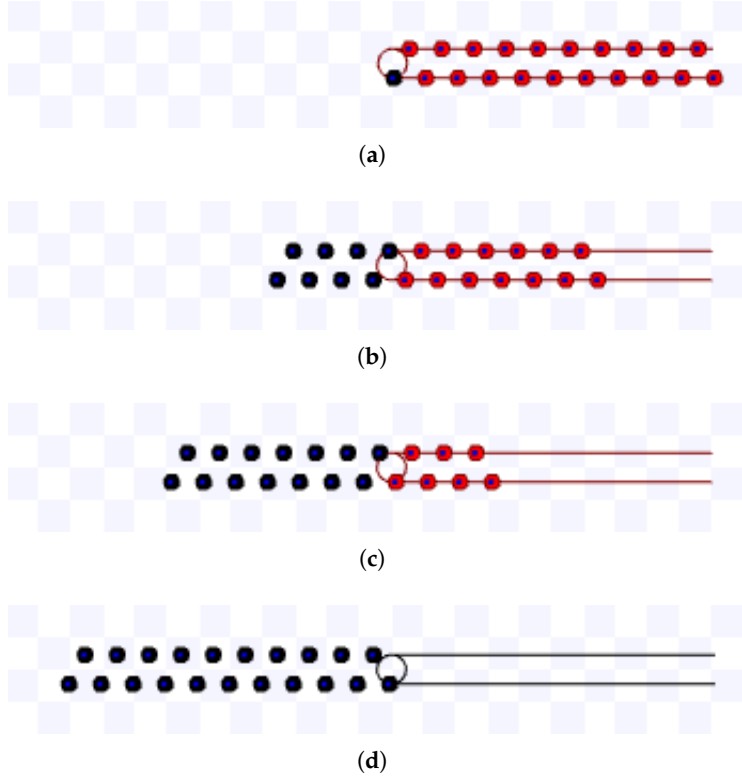

**Figure 21.** Simulation on Stage for compact lanes strategy using $s = 0.45$ m, $d = 1$ m during $T = 10.1$ s. Available on https://youtu.be/9OXGC1w83j0, accessed on 12 June 2022. (**a**) 0 s: beginning of the simulation; (**b**) After 3.5 s; (**c**) After 7 s; (**d**) 10.1 s: ending of the simulation.

Experiments were run in order to verify the throughput for a given time and the asymptotic throughput calculated by (2) to (5). Figure 22 shows the throughput for different values of time obtained by the experiments in Stage, i.e., $(N-1)/T$, in comparison with the calculated value by (2) and (3) for $s = 0.3$ m and by (4) and (5) for $s = 0.45$ m. "Simulation" stands for the data obtained from Stage, "Instantaneous" for the equations of the throughput for a given time calculated in (2) and (4) and "Asymptotic" for the asymptotic throughput obtained from (3) and (5). The mentioned results of the equations match the data obtained from simulations. These figures confirm that the equations presented in the theoretical section agree with the throughput obtained by simulations.

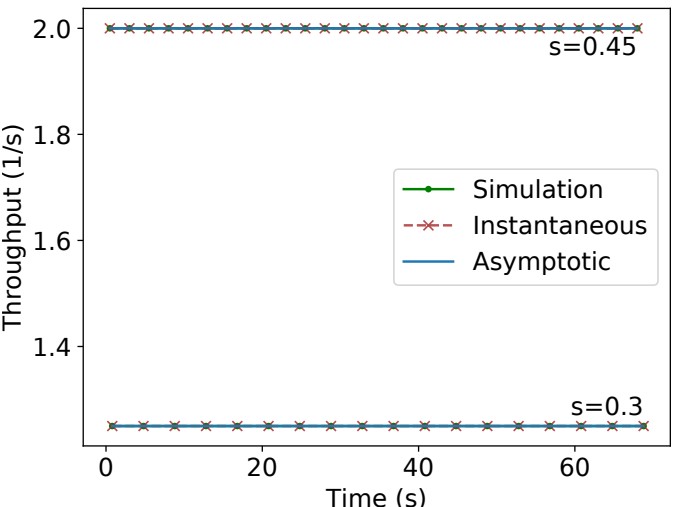

**Figure 22.** Throughput versus time plot for compact lanes strategy for different values of $s$.

### 4.2. Parallel Lanes

The parallel lanes strategy was experimented for $v = 1$ m/s and $s \in \{3, 6\}$ m. Figures 23 and 24 present screenshots from executions using these parameters.

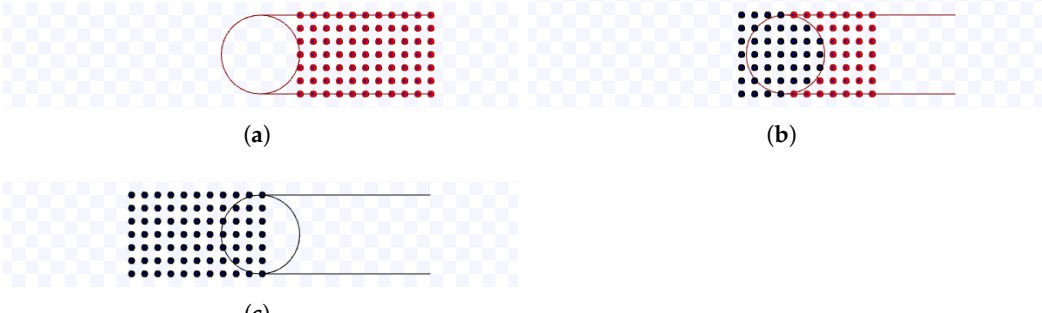

**Figure 23.** Simulation on Stage for parallel lanes strategy using $s = 3$ m, $d = 1$ m during $T = 13$ s. Available on https://youtu.be/2Y1RHc9YVaw, accessed on 12 June 2022. (**a**) 0 s: beginning of the simulation; (**b**) After 6.5 s; (**c**) 13 s: ending of the simulation.

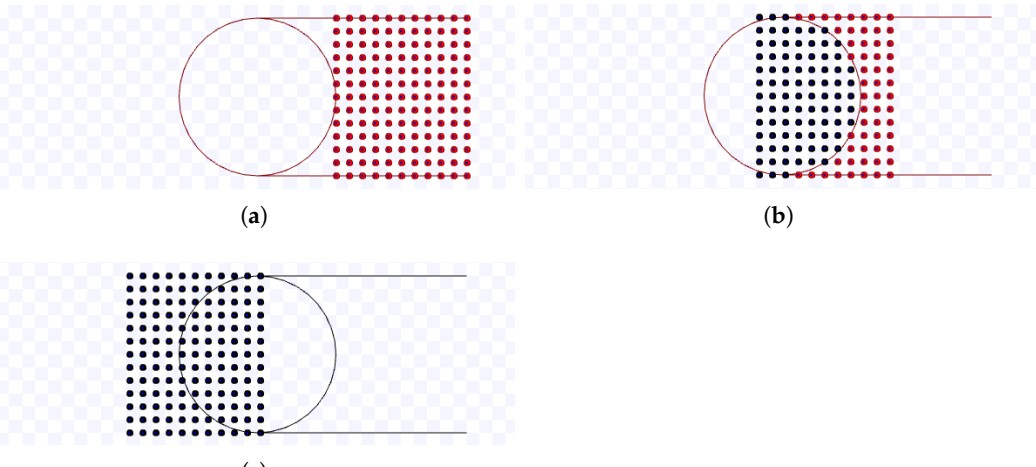

(a)

(b)

(c)

**Figure 24.** Simulation on Stage for parallel lanes strategy using $s = 6$ m, $d = 1$ m during $T = 16$ s. Available on https://youtu.be/TVdka65fi1g, accessed on 12 June 2022. (**a**) 0 s: beginning of the simulation; (**b**) After 8 s; (**c**) 16 s: ending of the simulation.

To verify the throughput for a given time calculated by (6) and its asymptotic value as in (7), they are compared with the throughput obtained from Stage simulations. Figure 25a presents these comparisons. "Simulation" stands for the data obtained from Stage, "Instantaneous" for the equations of the throughput for a given time calculated in (6), and "Asymptotic" for the asymptotic throughput obtained from (7). As expected, the values of (6) approximate to (7) as time passes. Additionally, observe that the values from (6) are almost aligned with the values from the simulation, except for some points. The difference in those points is due to the floating-point error discussed at the beginning of Section 4 that happens in the division before the use of floor or ceiling functions used on (6). Figure 25b shows the number of robots versus the time of arrival of the last robot for the same data used in Figure 25a. As the running time is proportional to the number of robots in the experiments, observe that the higher throughput per time is reflected as a lower arrival time of the last robot per the number of robots. In addition, note that the values tend to infinity as the horizontal axis values grow.

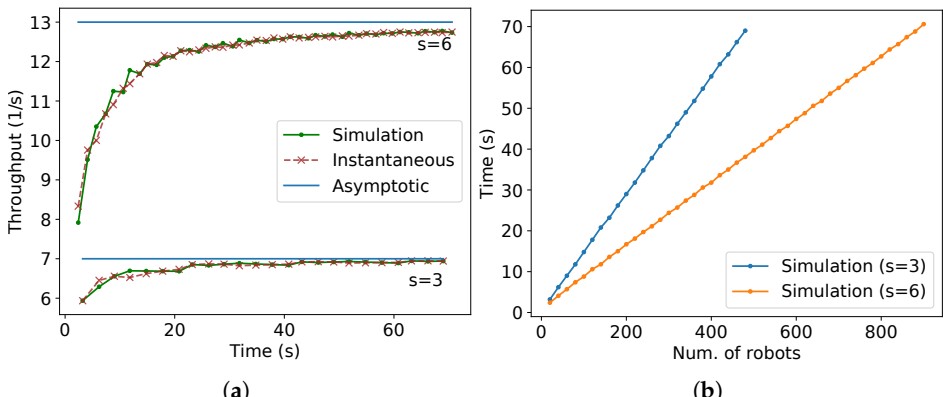

(a)

(b)

**Figure 25.** Plots for the experiments of parallel lanes strategy for $s \in \{3, 6\}$ m. (**a**) Number of robots versus throughput. (**b**) Number of robots versus the time of arrival of the last robot.

### 4.3. Hexagonal Packing

The hexagonal packing was experimented for $v = 1$ m/s and the combination of the following variables and values: $s \in \{3, 6\}$ m and $\theta \in \{0, \pi/12, \pi/6, 5\pi/18\}$. Figures 26–33 present screenshots from executions using these parameters.

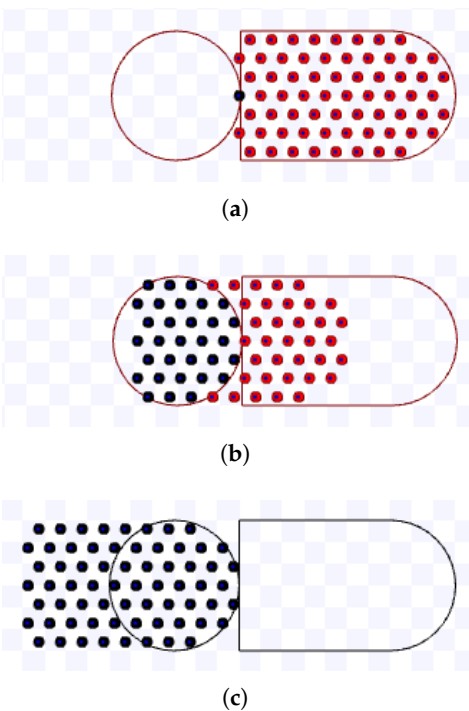

(a)

(b)

(c)

**Figure 26.** Simulation on Stage for hexagonal packing strategy using $s = 3$ m, $\theta = 0$ during $T = 9.8$ s. Available on https://youtu.be/6_LgZWFOWd0, accessed on 12 June 2022. (**a**) 0 s: beginning of the simulation; (**b**) After 4.9 s; (**c**) 9.8 s: ending of the simulation.

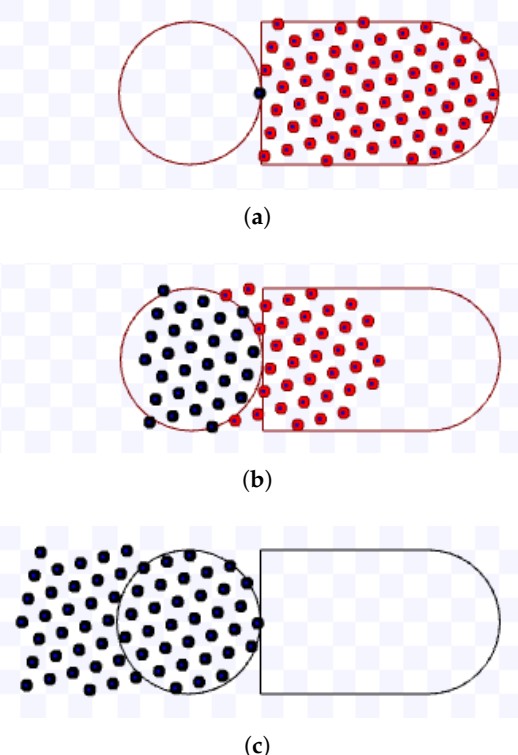

(a)

(b)

(c)

**Figure 27.** Simulation on Stage for hexagonal packing strategy using $s = 3$ m, $\theta = \pi/12$ during $T = 10$ s. Available on https://youtu.be/Wji8XlSQJBQ, accessed on 12 June 2022. (**a**) 0 s: beginning of the simulation; (**b**) After 5 s; (**c**) 10 s: ending of the simulation.

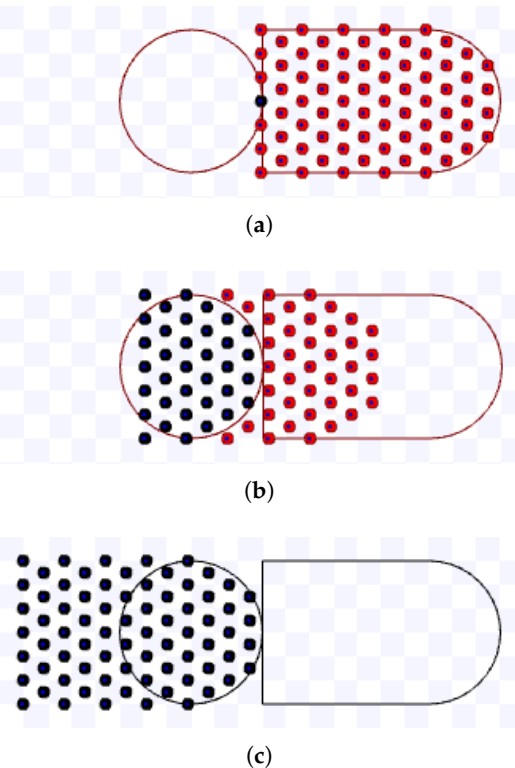

**Figure 28.** Simulation on Stage for hexagonal packing strategy using $s = 3$ m, $\theta = \pi/6$ during $T = 10$ s. Available on https://youtu.be/szOBU8no_sU, accessed on 12 June 2022. (**a**) 0 s: beginning of the simulation; (**b**) After 4.9 s; (**c**) 10 s: ending of the simulation.

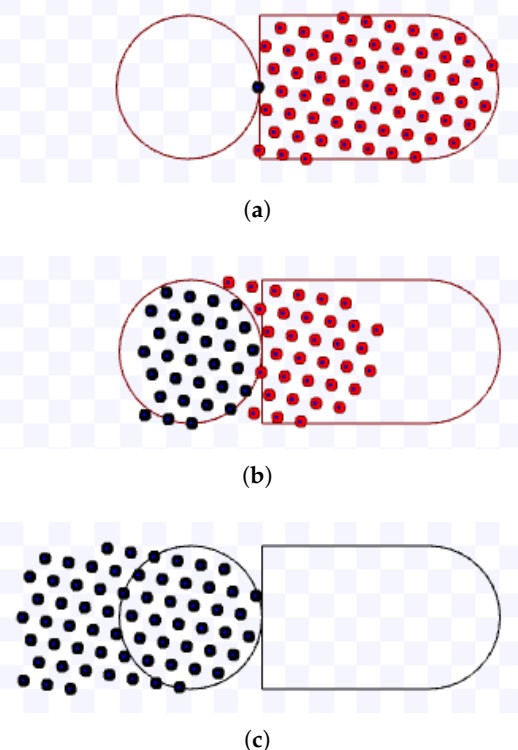

**Figure 29.** Simulation on Stage for hexagonal packing strategy using $s = 3$ m, $\theta = 5\pi/18$ during $T = 10$ s. Available on https://youtu.be/jRLgaF7Te1Q, accessed on 12 June 2022. (**a**) 0 s: beginning of the simulation; (**b**) After 4.9 s; (**c**) 10 s: ending of the simulation.

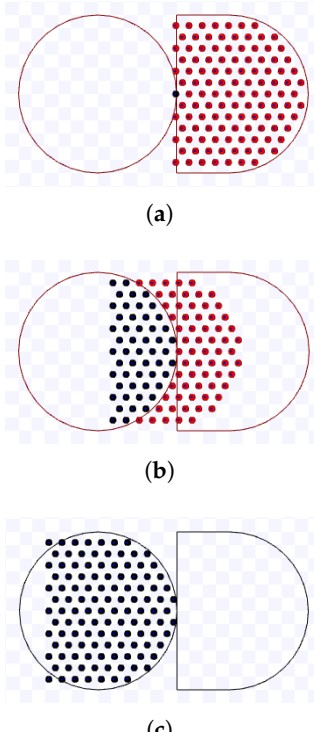

**Figure 30.** Simulation on Stage for hexagonal packing strategy using $s = 6$ m, $\theta = 0$ during $T = 9.8$ s. Available on https://youtu.be/v0FK8YpGrL8, accessed on 12 June 2022. (**a**) 0 s: beginning of the simulation; (**b**) After 4.9 s; (**c**) 9.8 s: ending of the simulation.

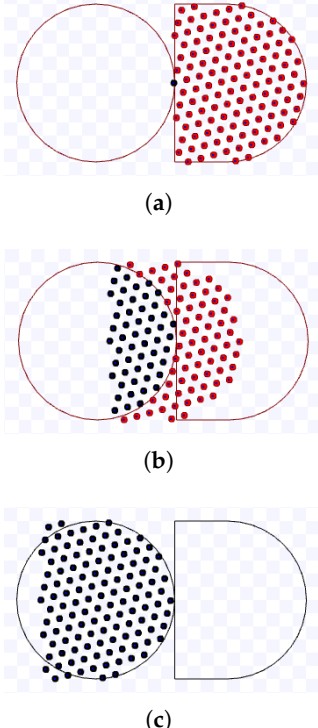

**Figure 31.** Simulation on Stage for hexagonal packing strategy using $s = 6$ m, $\theta = \pi/12$ during $T = 10.1$ s. Available on https://youtu.be/OBS_HADH5OE, accessed on 12 June 2022. (**a**) 0 s: beginning of the simulation.; (**b**) After 5 s; (**c**) 10.1 s: ending of the simulation..

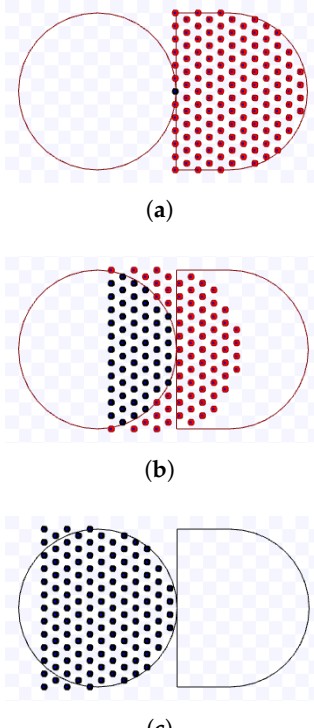

**Figure 32.** Simulation on Stage for hexagonal packing strategy using $s = 6$ m, $\theta = \pi/6$ during $T = 10$ s. Available on https://youtu.be/-KX7ziOp8b0, accessed on 12 June 2022. (**a**) 0 s: beginning of the simulation.; (**b**) After 4.9 s; (**c**) 10 s: ending of the simulation.

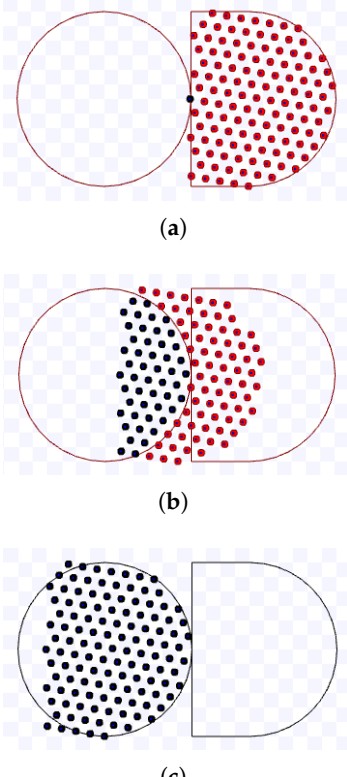

**Figure 33.** Simulation on Stage for hexagonal packing strategy using $s = 6$ m, $\theta = 5\pi/18$ during $T = 10$ s. Available on https://youtu.be/GRYRnH5CrhU, accessed on 12 June 2022. (**a**) 0 s: beginning of the simulation; (**b**) After 4.9 s; (**c**) 10 s: ending of the simulation.

To evaluate the throughput for a given time and angle calculated in (9) and the bounds on the asymptotic throughput as in (11), they are compared with the throughput obtained from Stage simulations. Figure 34 presents these comparisons. Observe that the values from (9) are almost aligned with the values from the simulation, except for some points. The difference in those points is also due to the floating-point error—discussed in the introduction of Section 4—over the divisions and trigonometric functions performed before the use of floor or ceiling functions used on (9). In addition, due to the floating-point error, in the computation of (10), instead of using $\min(L(x_h), C_2(x_h)) = \lfloor L(x_h) \rfloor$, $|\min(L(x_h), C_2(x_h)) - \lfloor L(x_h) \rfloor| < 0.001$ was checked.

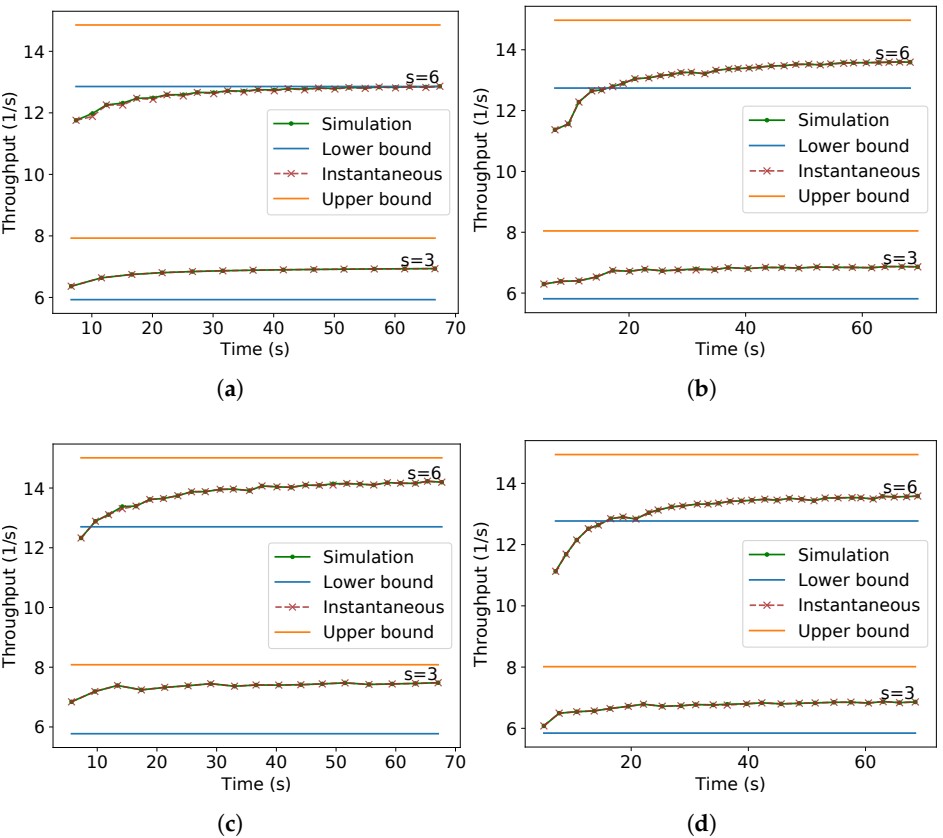

**Figure 34.** Comparison of simulation data with the asymptotic and instantaneous throughput for hexagonal packing with different values of $s$ and $\theta$. (**a**) $\theta = 0$; (**b**) $\theta = \pi/12$; (**c**) $\theta = \pi/6$; (**d**) $\theta = 5\pi/18$.

Additionally, note in Figure 34 that for any value of $s$ or $\theta$, as the time passes, the values of (9) asymptotically approach some value inside the bounds given by (11). Although the exact asymptotic value could not be given for the presented parameters, the experiments show that the bounds are correct. In the same manner, as occurred for parallel lanes, the higher throughput per time is reflected as a lower arrival time of the last robot per the number of robots, and it tends to infinity as the number of robots grows (Figure 35).

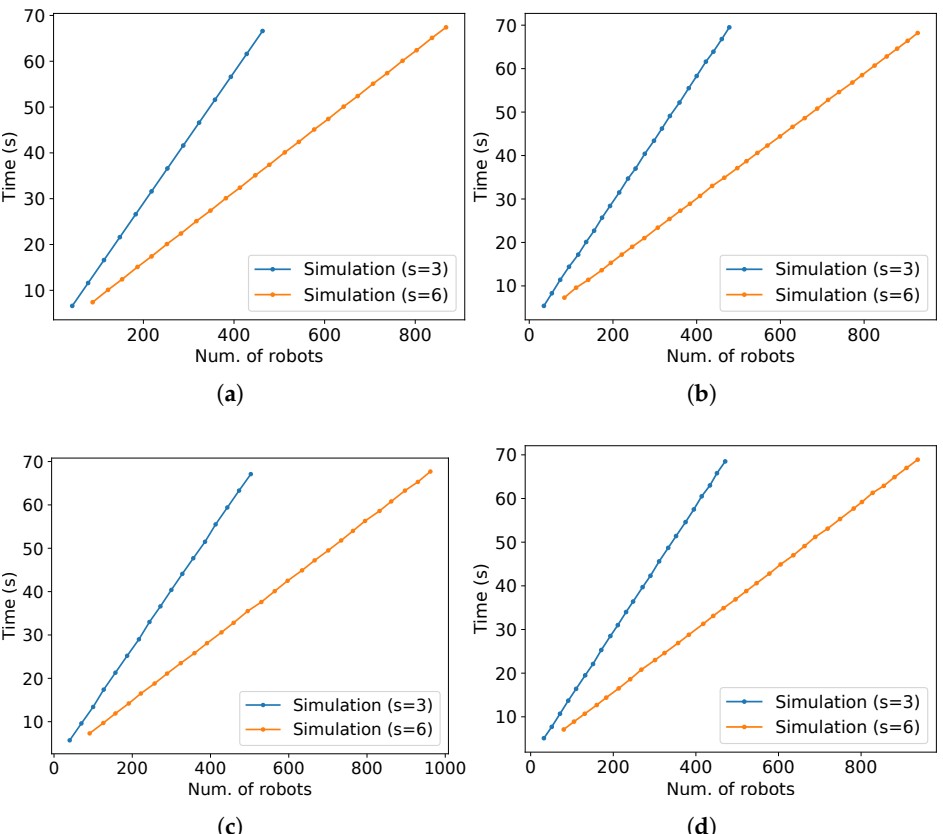

**Figure 35.** Time of arrival at the target of the last robot versus the number of robots for the same simulations in Figure 34. (**a**) $\theta = 0$; (**b**) $\theta = \pi/12$; (**c**) $\theta = \pi/6$; (**d**) $\theta = 5\pi/18$.

*4.4. Touch and Run*

For the touch and run strategy, the robots maintain the linear speed over the whole experiment, then turn at a fixed constant rotational speed $\omega = v/r$, for $r$ obtained from (13), when they are next to the target centre by the distance $d_r$ obtained from (12). After they arrive at the target region, when they are distant from the target centre by $d_r$, they leave the curved path, stop turning and follow the linear exiting lane. On that lane, to stabilise their path following, the robots follow the queue using a turning speed equal to $\gamma - \beta$, such that $\beta$ is the angle of the exit lane and $\gamma$ is the robot orientation angle, both in relation to the $x$-axis.

The speed of these experiments was $v = 0.1$ m/s because the robots utilised on Stage have a maximum turning speed of $\pi/2$ rad/s. Choosing a low linear speed implies a greater number of lanes $K$, as the turning speed $\omega = v/r$ and $r$ vary over $K$ and $s$. In addition, a low linear speed diminishes the time measurement error, since the positions of the robots are sampled at every 0.1 s by the Stage simulator. Their positions are not guaranteed to be obtained at the exact moment they are far from the target centre by $d_r$; thus, this also yields an error in time measurement for their arrival in the target area.

The value of $s$ is in $\{3, 6\}$ m and all allowed $K$ values are used for experimenting with the touch and run strategy with 200 robots. By (14), for the former $s$ value, there is a maximum $K = 18$ and for the later, $K = 37$. However, as the maximum angular speed is limited, the allowed $K$ values range for $s = 3$ m is reduced to $\{3, \ldots, 16\}$ and for $s = 6$ m, $\{3, \ldots, 33\}$. Figures 36–39 present screenshots from executions using some of these parameters. The circle in the middle of these figures is the target region, and the lines where the robots are over represent the curved trajectory they follow by the touch and run strategy.

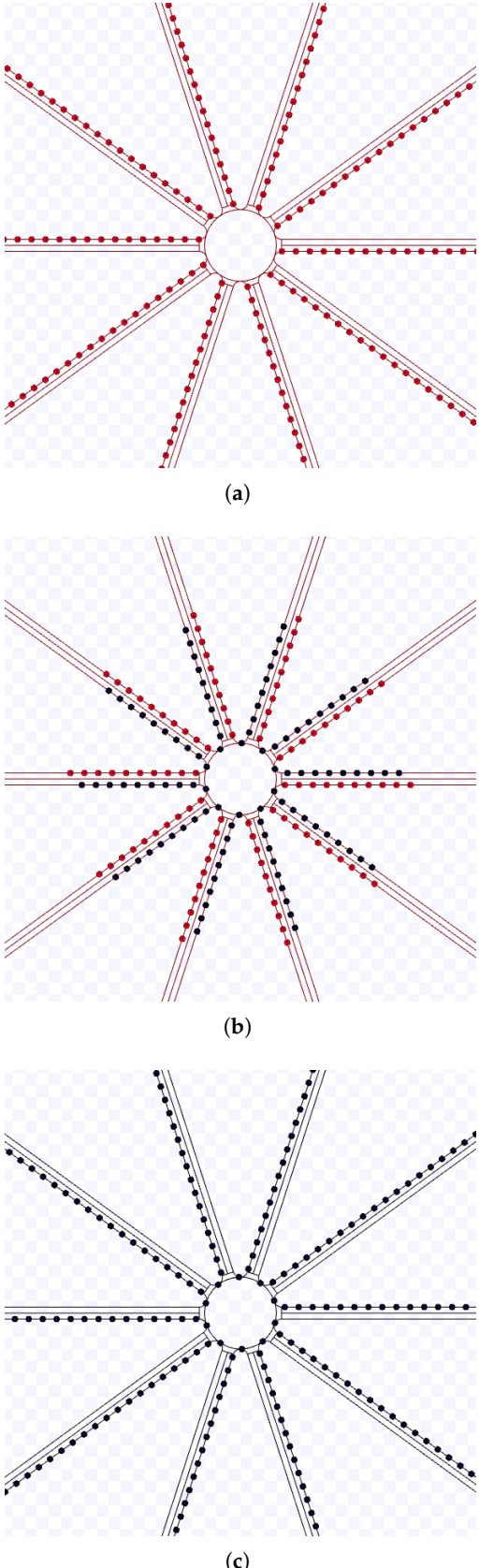

**Figure 36.** Simulation on Stage for the touch and run strategy using $s = 3$ m, $K = 10$ during $T = 228$ s at $v = 0.1$ m/s. Available on https://youtu.be/Z-ruOMYFyBU, accessed on 12 June 2022. (**a**) 0 s: beginning of the simulation; (**b**) After 114 s; (**c**) 228 s: ending of the simulation.

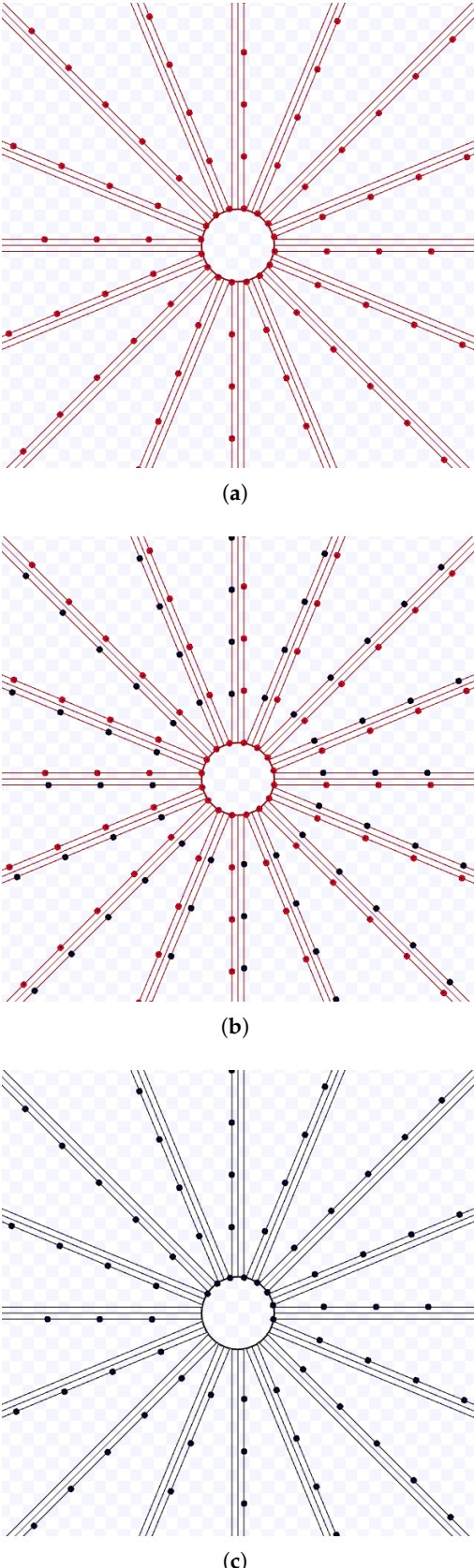

**Figure 37.** Simulation on Stage for the touch and run strategy using $s = 3$ m, $K = 16$ during $T = 523.1$ s at $v = 0.1$ m/s. Available on https://youtu.be/FvAqv0zD4_Y, accessed on 12 June 2022. (**a**) 0 s: beginning of the simulation; (**b**) After 261.6 s; (**c**) 523.1 s: ending of the simulation.

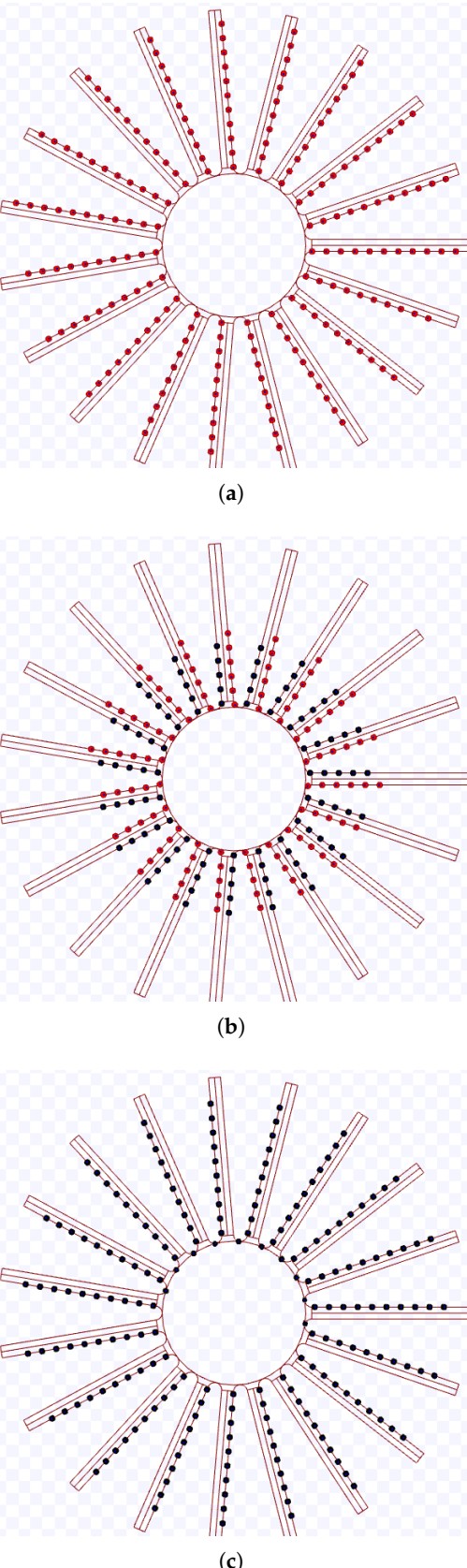

**Figure 38.** Simulation on Stage for the touch and run strategy using $s = 6$ m, $K = 19$ during $T = 127.4$ s at $v = 0.1$ m/s. Available on https://youtu.be/xJVoVCIjX5k, accessed on 12 June 2022. (**a**) 0 s: beginning of the simulation; (**b**) After 63.6 s; (**c**) 127.4 s: ending of the simulation.

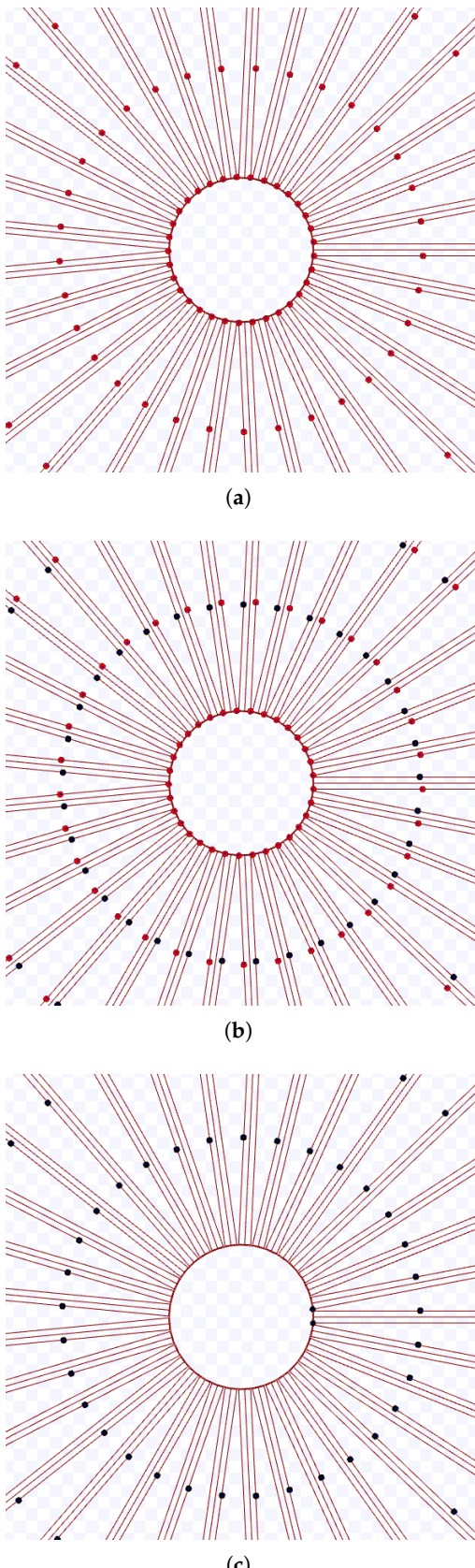

**Figure 39.** Simulation on Stage for the touch and run strategy using $s = 6$ m, $K = 33$ during $T = 548$ s at $v = 0.1$ m/s. Available on https://youtu.be/-xZz84npKV4, accessed on 12 June 2022. (**a**) 0 s: beginning of the simulation; (**b**) After 274 s; (**c**) 548 s: ending of the simulation.

Figure 40 presents the comparison of (16) and (19) for the throughput for a given time, the bound on its asymptotic value and the one obtained from Stage simulations. Although the total number of robots and the linear speed were fixed, the arrival times and the number of robots to reach the target change for each parameter used in this figure since the distance between the robots per lane varies and the number of robots simultaneously arriving is, in most cases, the number of lanes. In addition, the first two arrival times were not plotted because the first one is zero, yielding an indeterminate output by the throughput definition, and the second one is still too small in relation to the others, making the resultant throughput too high compared with the rest, thus producing an incomprehensible graph.

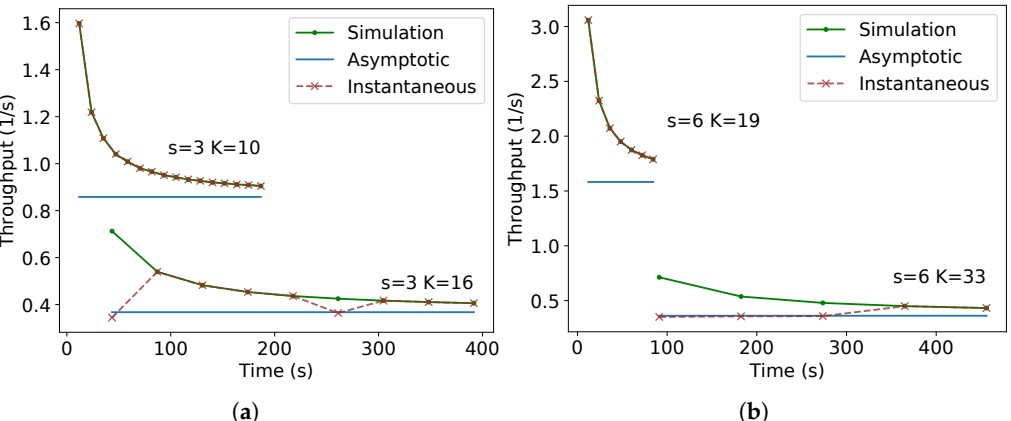

**Figure 40.** Throughput versus time comparison of the touch and run simulation on Stage with asymptotic values and the theoretical instantaneous equation for the throughput for different values of $s$ and $K$. (**a**) $s = 3$ m and $K \in \{10, 16\}$; (**b**) $s = 6$ m and $K \in \{19, 33\}$.

Observe that the values from (16) are almost equal to the values from simulation, except for some points. They are different because of the floating-point error in the divisions and trigonometric functions before the use of floor function used on (16)—already mentioned in the introduction of Section 4—as well as the time measurement errors for the arrival of the robots on the target area as explained at the beginning of this section. As expected, the values of (16) tend to come nearer to the asymptotic value given by (19). Differently from the previous strategies, notice that, for small values of $T$, the throughput is higher than for larger ones because, for a fixed $K$, (16) is decreasing for $T$. As occurred for the previous strategies, higher throughput per time is reflected as a lower arrival time of the last robot per number of robots, which tends to infinity as the number of robots grows (Figure 41).

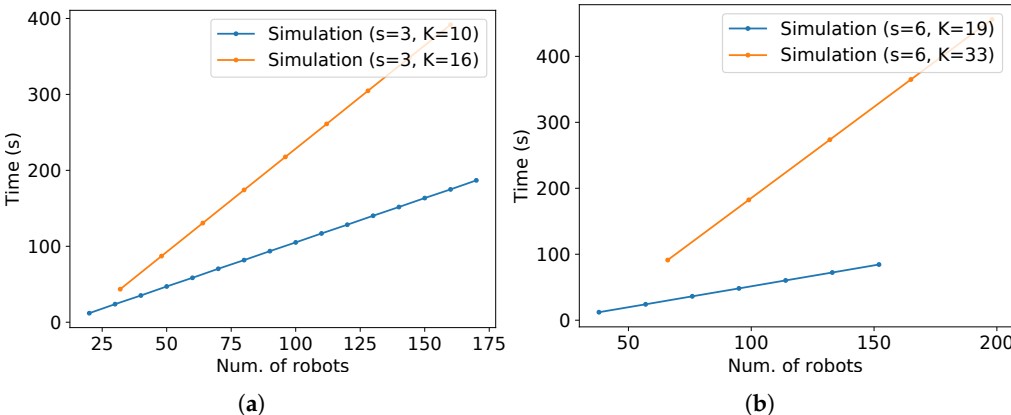

**Figure 41.** Time of arrival at the target of the last robot versus number of robots for the same simulations in Figure 40. (**a**) $s = 3$ m and $K \in \{10, 16\}$; (**b**) $s = 6$ m and $K \in \{19, 33\}$.

Figure 42 shows a comparison of the throughput at the end of the experiment—that is, for 200 robots and considering the difference between the time to reach the target region spent by the last robot and the first—and the asymptotic throughput obtained by (19) for all the possible number of lanes (*K*) for the used parameters and $s \in \{3, 6\}$ m. The simulation values tend to come close to the asymptotic value, confirming the theoretical results.

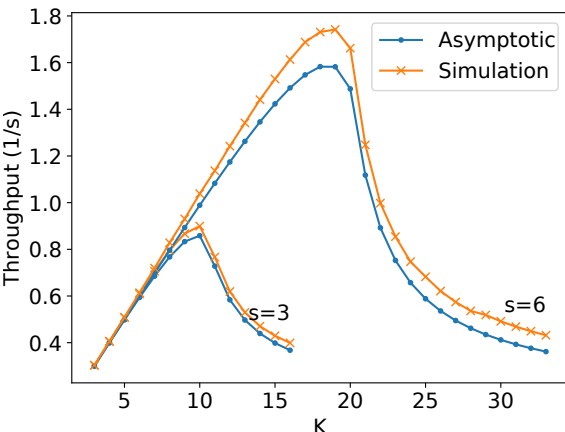

**Figure 42.** Throughput versus the number of lanes comparison of the simulation on Stage and asymptotic throughput for $s \in \{3, 6\}$ m.

### 4.5. Comparison between Hexagonal Packing and Parallel Lanes

As discussed in Section 3.3.4, it is observed that the parallel lane strategy has a higher throughput than hexagonal packing for values of $u = s/d$ from 0.5 to a value of about 0.85 and for high values of $T$, despite the parallel lanes having lower asymptotic throughput for other values of $u$. In order to validate this observation, experiments on Stage were performed for these strategies using $T = 10,000$ s, $v = 0.1$ m/s, $d = 1$ m and $s$ ranging from 0.4 to 0.95 m in increments of 0.05 m. The best hexagonal packing angle $\theta$ was computed for hexagonal packing using the same method mentioned at the end of the theoretical section, i.e., the maximum throughput was searched using 1000 evenly spaced points between $[0, \pi/3)$ to find the best $\theta$; then, it was compared with the result for $\pi/6$.

Figure 43 presents the results from the experiments with Stage and the theoretical results shown earlier. The functions $f_h$ and $f_p$ are the same presented in Figure 17. The labels "Simulation hex." and "Simulation par." stand for the throughput resultant from the experiments with hexagonal packing and parallel lanes strategies, respectively. The throughput improvement for the values of $u = s/d$ where the parallel lanes strategy overcomes the hexagonal packing is mainly caused by the square packing being more effective than hexagonal packing for fitting the robots inside the circle over the time for those values. To illustrate this, Figure 44 illustrates the execution for $v = 0.1$ m/s, $d = 1$ m and $s \in \{0.5, 0.85\}$ m. The robots run from right to left at a constant linear speed $v = 0.1$ m/s. The grey squares are highlighted—which measure $1 \times 1$ m²—to help estimate the time needed for about eight robots to arrive in the target region. This figure shows that the square packing fits more robots than hexagonal packing over time in these cases.

Observe in these figures that when the robots are arranged in squares, more robots arrive per unit of time than using hexagonal packing. To help visualise this, heed that in Figure 44a, there are $N = 9$ robots in black, occupying a rectangle including the circular target area with a width of approximately $W \approx 4.5$ m (this distance can be roughly measured by the grey squares, counting from the two last black robots on the right side to the first one in the left side). As $v = 0.1$ m/s was assumed, the throughput in this case is approximately $\frac{N-1}{\frac{W}{v}} \approx \frac{(9-1)0.1}{4.5} \approx 0.178$ s⁻¹. Making similar calculations, Figure 44b–d have the approximate throughputs $\frac{(8-1)0.1}{3} \approx 0.233$ s⁻¹, $\frac{(8-1)0.1}{4} = 0.175$ s⁻¹ and $\frac{(8-1)0.1}{3} \approx 0.233$ s⁻¹, respectively. The results from the parallel lanes in this illustration—about 0.233 for both values of $s$—surpass the values for the hexagonal packing.

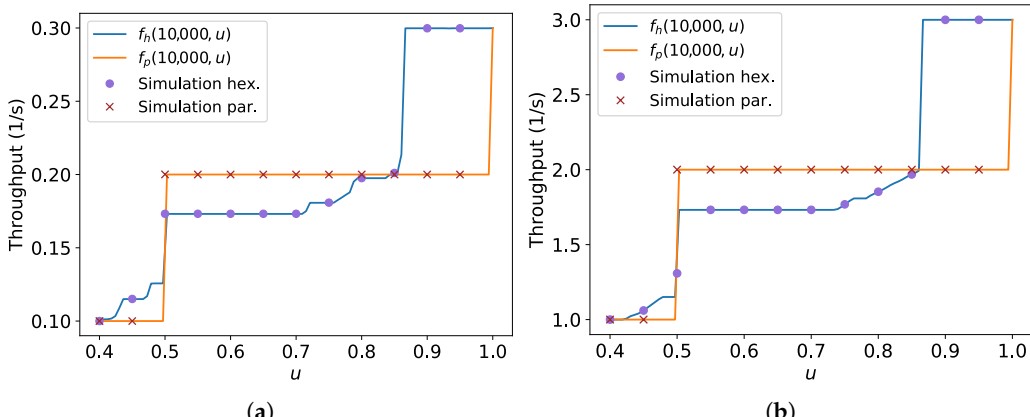

(a)　　　　　　　　　　　　　(b)

**Figure 43.** Throughput versus ratio $u = s/d$ comparing hexagonal packing and parallel lanes strategies for $v \in \{0.1, 1\}$ m/s, including results from Stage simulations. (**a**) $v = 0.1$ m/s; (**b**) $v = 1$ m/s.

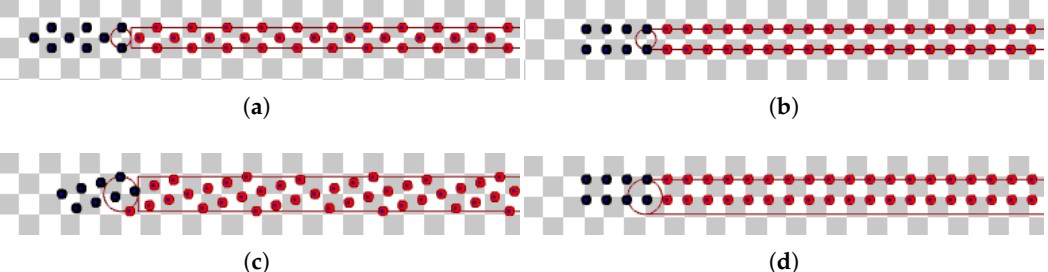

(a)　　　　　　　　　　　　　(b)

(c)　　　　　　　　　　　　　(d)

**Figure 44.** Screenshots of the Stage simulation for hexagonal packing and parallel lanes strategy for $d = 1$ m and $s \in \{0.5, 0.85\}$ m. (**a**) Hexagonal packing with best $\theta$ for $s = 0.5$ m. Available on https://youtu.be/IZBnFHLKXUA, accessed on 12 June 2022. (**b**) Parallel lanes for $s = 0.5$ m. Available on https://youtu.be/YYv1dJFkdPA, accessed on 12 June 2022. (**c**) Hexagonal packing with best $\theta$ for $s = 0.85$ m. Available on https://youtu.be/r9X0fsnngm0, accessed on 12 June 2022. (**d**) Parallel lanes for $s = 0.85$ m. Available on https://youtu.be/0cx-bHPIong, accessed on 12 June 2022.

## 5. Conclusions

A novel metric was proposed for measuring the effectiveness of algorithms to minimise congestion in a swarm of robots trying to reach the same goal: the common target area throughput. In addition, the asymptotic throughput for the common target area was defined as the throughput when the time tends to infinity.

Assuming that the robots move at constant maximum speed and the distance between each other is as close as possible to a fixed value, it was shown how to calculate the maximum throughput for different theoretical strategies to arrive at the common circular target region: (i) making parallel queues to reach the target region, (ii) using a corridor with robots in hexagonal packing to enter in the region, and (iii) following curved trajectories to touch the region boundary. Based on these calculations solely, it was possible to compare which strategy is better.

Due to their aim of maximising the target area throughput, these strategies were used as inspiration for new algorithms using artificial potential fields in [38]. Thus, for common target area congestion in robotic swarms, the throughput is well suited for comparing algorithms due to its abstraction of the rate of the target area access as the number of robots grows, whether the closed throughput equation is derived or not.

The key contribution of this work is a fundamental theoretical study of congestion in swarm robotics, which already served as inspiration to create new algorithms. However, future work could extend this theoretical study further, by considering varying linear speed and distances between the robots.

**Supplementary Materials:** The following supporting information can be downloaded at: https://www.mdpi.com/article/10.3390/math10142482/s1. References [39–41] are cited in the supplementary materials.

**Author Contributions:** Conceptualization, X.D.; Formal analysis, Y.T.d.P. and X.D.; Investigation, Y.T.d.P. and X.D.; Methodology, Y.T.d.P. and X.D.; Software, Y.T.d.P. and X.D.; Supervision, L.S.M.; Writing—original draft, X.D.; Writing—review & editing, Y.T.d.P. and L.S.M. All authors have read and agreed to the published version of the manuscript.

**Funding:** This research was funded by Lancaster University.

**Acknowledgments:** Yuri Tavares dos Passos gratefully acknowledges the Faculty of Science and Technology at Lancaster University for the scholarship and the Universidade Federal do Recôncavo da Bahia for granting the leave of absence to finish his PhD.

**Conflicts of Interest:** The authors declare no conflict of interest.

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
