# Peer review of "On the Throughput of the Common Target Area for Robotic Swarm Strategies"

_mathematics, doi:10.3390/math10142482_

Round 1
Reviewer 1 Report
The authors proposed the target area throughput and its asymptotic throughput factors for evaluating the access efficiency of a common target area as the number of robots in the swarm is increased. The manuscript includes many equations, figures, definitions, propositions, proofs, and lemma, so it is prolonged. Thus, I suggest either dividing the manuscript into two parts (two manuscripts) or removing some of the proofs and representing them as supplementary files. However, the manuscript is presented in a well way. In the following, there are many comments; I think they are useful to enhance the manuscript.
1. Please, do not use pronouns such as; “we” in lines 2, 5, 6, 9, 35, 37, 42, 45, 56, 57, 58, 59, 60, 64, 65, 67, 69, 71, 75, 77, …etc. and “our” in line 11. I think the authors used the pronoun “we” around 466 times in the whole manuscript, it is too much. It is preferred to use the passive voice format in scientific writing.
2. I suggest supporting the abstract section with numerical results to show the superiority of the proposed strategies for alleviating congestion of the robotic swarm in a circular target area.
3. The captions of Figures 7, 9, 10, 11, 12, 14, 18, 19, 20, 21, 30, 33, 42, 51, and 52 are very prolonged. Please, revise them to be shorter by describing the figures in the text.
4. Justify why did the authors select 32 robots in Figure 11?
5. Justify why did the authors use “99 equally spaced values for θ ∈ [0, π/3) on the left-hand side images and 100 on the right-hand side”, meanwhile, they chose 10^7 and 10^7 + 1 equally spaced in Figures 16 and 17.
6. Please, revise the caption of Figures 15 and 17.
7. The conclusion is prolonged. Please, revise it by focusing only on the significant outcomes of the manuscript.
8. The manuscript needs language polishing because it has some grammatical mistakes and typos.
Author Response
Reply about dividing the paper: Other reviewers suggested moving most of the technical details to the Online Appendix to improve readability. Following that, the main paper now has 33 pages.
Reply about point 2:
This paper aims to show measures for comparison of different strategies for the congestion problem. The key contribution of this work is not the proposal of new algorithms to alleviate congestion but a fundamental theoretical study of the congestion problem in swarms having the same target. In our theoretical strategies, we assume robots move at maximum speed because we derive the maximum possible throughput. Hence, that would not be directly comparable numerically with existing robotic algorithms for swarm congestion since robots would move with varying speeds. However, our presented strategies serve as inspiration for the development of novel algorithms, which we present in [27 in first version]. Additionally, they give theoretical bounds for the expected throughput in real situations.
Reply about point 4: This was meant to be an example. I edited it so that it is explicitly written in the paragraph explaining the Figure A5 starting at line 162 in the Online Appendix (this figure had the number 11 in the first version, but it is now in the Online Appendix).
The suggestions 1, 3, 5, 6, 7 and 8 were performed as described.
Thank you for all your commendations and suggestions. They were all welcome.
Reviewer 2 Report
This paper focuses on a fundamental theoretical study of the congestion problem in swarm robotics when the target area is shared, where a method for evaluating algorithms for the common target problem in a robotic swarm by using the throughput in theoretical or experimental scenarios is proposed.
Overall, the paper is clearly organized and the results are interesting. I would like to provide the following revision points:
(1) The original papers of main lemmas are better to be provided.
(2) Why only the case of Proposition 2 and 3 can consider the maximum speed, but other results consider a constant speed? The case with dynamic speed and inter-robot distance is important in practice, which may have great influence on the throughput.
(3) Additional contents in Section 3 can be kindly reduced to improve the readability of the paper.
(4) Kindly enrich the literature review of this paper by citing additional references related to this topic.
(5) Many illustration and simulation figures are given, so the manuscript should be formatted better for readability.
Author Response
Reply to 1: All the proofs of the presented lemmas are in the Online Appendix. This is mentioned in the last paragraph of the introduction of Section 3 (starting at line 188 in the new version). These are not inside the main paper to reduce the number of pages. Also, other reviewers suggested moving most of the technical details to an appendix to improve readability.
Reply to 2: I corrected these assumptions. They were meant to be constant, but at the maximum. When we assume that the robots are constantly moving at maximum linear speed and maintaining a fixed minimum distance, we can provide analytical calculations of the maximum possible throughput for a given time and bounds or exact value of the maximum asymptotic throughput for the different theoretical strategies. Based solely on these calculations, we can compare which strategy is better. However, for robots using artificial potential fields, it is not straightforward to get explicit throughput equations due to the changeability of those quantities previously assumed constant. Then, in the lack of closed asymptotic equations, simulations were performed in [27 in first version] for the algorithms inspired by our strategies in order to get experimental throughput and compare algorithms for varying linear speed and inter-robot distances. As shown by that experimental data, their variation and the effect of the other robots in the trajectory does affect the throughput. However, the analytically calculated maximum throughput in this work serves as an upper bound to the ones obtained from the simulations in more realistic conditions when considering the mean speed and mean distance between the robots in place of the constant values on the obtained equations. This justification is now in the penultimate paragraph of the introduction (starting at line 101). Additionally, we edited the list of contributions in the introduction (starting at line 87), and the second paragraph after the Definition 2 in Section 3 (starting at line 168).
Reply to 3: All the proofs were put in the Online Appendix, reducing Section 3.
Reply to 4: The introduction was now modified to include eleven new references.
Reply to 5: After the last modifications in the text, the figures were put next to the text where they are invoked. A few were reduced in scale, but we prioritised not hindering the visualisation of the letters or other details of the figure. If we put the figures only before the invoking text, lots of blank space without text would be generated because the size of the figures (mainly in the simulation section) is greater than the text discussing them. Thus, to avoid the increase in the page number and smartly use the paper space, we preferred to present the figures as soon as possible and write text in the blank spaces that appear.
Thank you for all your commendations and suggestions. They were all welcome.
Reviewer 3 Report
Please see the enclosed review.

Author Response
The shortening of Section 3 was also asked by other reviewers, so a rearrangement was performed such that the proofs were all put in the Online Appendix. Figures 7, 9 and 10 were moved to the Online Appendix, as they were intended to aid the understanding of the proof where they appear. However, the figures illustrating examples of the strategies were maintained in the main paper.
Thank you for all your commendations and suggestions. They were all welcome.
Round 2
Reviewer 3 Report
The authors took into account the reviewers' comments and the paper was considerably improved. The paper is ready for publication.